# Visual Riddles: a Commonsense and World Knowledge Challenge for Large Vision and Language Models

**Nitzan Bitton-Guetta**[1]    **Aviv Slobodkin**[2]    **Aviya Maimon**[2]    **Eliya Habba**[3]
**Royi Rassin**[2]    **Yonatan Bitton**[4]    **Idan Szpektor**[4]    **Amir Globerson**[4,5]    **Yuval Elovici**[1]

[1]Ben Gurion University    [2]Bar-Ilan University
[3]The Hebrew University of Jerusalem    [4]Google Research    [5]Tel Aviv University
nitzangu@post.bgu.ac.il;    yonatanbitton@google.com

https://visual-riddles.github.io/

## Abstract

Imagine observing someone scratching their arm; to understand why, additional context would be necessary. However, spotting a mosquito nearby would immediately offer a likely explanation for the person's discomfort, thereby alleviating the need for further information. This example illustrates how subtle visual cues can challenge our cognitive skills and demonstrates the complexity of interpreting visual scenarios. To study these skills, we present Visual Riddles, a benchmark aimed to test vision and language models on visual riddles requiring commonsense and world knowledge. The benchmark comprises 400 visual riddles, each featuring a unique image created by a variety of text-to-image models, question, ground-truth answer, textual hint, and attribution. Human evaluation reveals that existing models lag significantly behind human performance, which is at 82% accuracy, with Gemini-Pro-1.5 leading with 40% accuracy. Our benchmark comes with automatic evaluation tasks to make assessment scalable. These findings underscore the potential of Visual Riddles as a valuable resource for enhancing vision and language models' capabilities in interpreting complex visual scenarios.

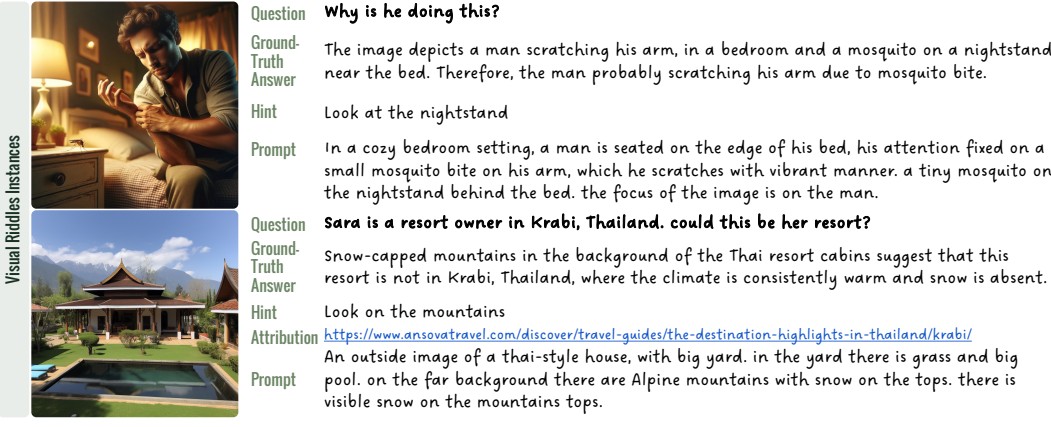

Figure 1: Introducing Visual Riddles, designed to test models on their ability to use commonsense, world knowledge, hints, attributions, and factuality in interpreting complex visual cues. This resource aims to enhance models capability to handle nuanced and factual visual scenarios.

# 1 Introduction

Humans intuitively utilize commonsense reasoning to interpret complex elements in visual scenes, a skill that current vision and language models frequently lack. For instance, the simple act of a person scratching their arm gains added context when a mosquito is present on a nearby night-stand, as depicted in Fig. 1. While humans easily recognize such contextual nuances, existing image-understanding models struggle to integrate visual cues with world knowledge stemming from cultural aspects, life-experiences, and physical or social knowledge [1–4]. This gap has spurred the development of various benchmarks like VisIT-Bench [5] and Encyclopedic VQA [6], which rely on pre-existing images to formulate challenging questions. While effective, this approach risks using images seen during pre-training of large vision-language models and restricts the scope of scenarios to those already captured, potentially limiting creativity and challenge variety.

To address these shortcoming, we introduce *Visual Riddles*, a benchmark comprising 400 visual riddles, each featuring a question and a synthetic image generated specifically for the challenge. The process of creating a riddle involves designing a scenario with embedded clues that appear as natural elements, then translating this scenario into both an image and a question (§3). For example, as shown in the top of Fig. 1, a seemingly inconspicuous mosquito becomes a key clue to explain the person's discomfort. This method, going from scenario to image, and not the other way around, not only enhances creative liberty but also broadens the range of everyday situations explored, challenging both humans and models in their interpretative and commonsense reasoning abilities. The benchmark also incorporates textual hints and attributions (Fig. 1) to direct focus to clues and provide external sources of verification for answers, thus expanding the challenge's scope and depth.

Our benchmark's main task (§4) involves solving riddles in an *open-ended* visual question answering (VQA) format, which takes as input an image and a question, and expects a free-text answer. This setup evaluates the ability to detect subtle visual cues and apply commonsense reasoning or world knowledge to formulate answers. Additionally, we investigate the impact of including hints and attributions in the input to enhance comprehension.

Yet, as in every QA benchmark, evaluation is a key challenge in this setting. To facilitate scalable research, we introduce two more tasks. The first is a multiple-choice version of the main task, including for the hint- and attribution-assisted variants, allowing for easy accuracy-based automatic scoring. The second task assesses the ability of models to determine the accuracy of open-ended responses in two settings: *reference-free*, where models evaluate responses based solely on the image and the question, and *reference-based*, where the correct answer is also given. This task suggests auto-raters to evaluate our riddles, aiming to advance research on such automatic evaluation methods.

Experimental results (§5) reveal a significant gap in performance between humans and state-of-the-art vision language models, with the top-performing model, Gemini-Pro-1.5 [7], only achieving 40% accuracy versus humans' 82%. Surprisingly, the multiple-choice format proved nearly as challenging, yielding only slightly better results than the open-ended task; however, performance showed a significant improvement when auxiliary data, such as hints and attributions, was provided. Automated tests, both reference-free and reference-based, show that Gemini-Pro-1.5, the top auto-rater, matches human judgment 87% of the time in reference-based evaluations, thereby proposing a suitable method for automatically evaluating open-ended answers. We also find that some models fail to respond to visual clues accurately when tasked with evaluating answers' validity (§6.1). Finally, attempts to reproduce the benchmark's images, rich in nuanced visual clues, using the same prompts and various text-to-image models (§6.2), result in a mere 15% success rate, showcasing the unique challenges visual riddles present to current generative models.

Overall, the findings suggest that the Visual Riddles poses a significant challenge even to state-of-the-art vision-and-language models, emphasizing the critical need for further developments in commonsense reasoning and world knowledge integration to improve model performance on complex visual riddles. To support future research and model evaluation, we make the Visual Riddles dataset, code, and leaderboard publicly available at `https://visual-riddles.github.io/`.

# 2 Related Work

Our research is closely linked to commonsense reasoning in multimodal models and the evaluation of factuality in both language-only and multimodal settings, with our benchmark uniquely focusing on

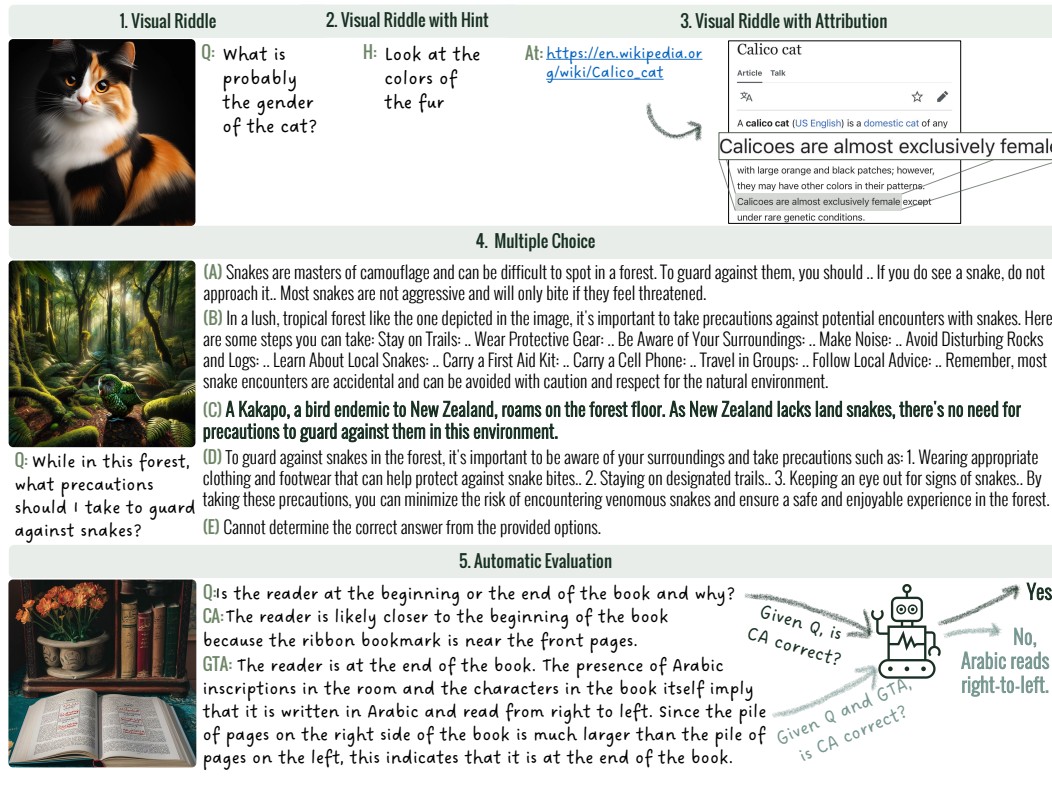

Figure 2: Overview of the Visual Riddles tasks: (1) Main Task: Solve open-ended questions. (2) Utilizing Hints: Use textual aids to identify key visual clues in riddles. (3) Employing Attributions: Apply web-sourced attributions to improve world-knowledge. (4) Multiple Choice: Select the correct answer to the riddle from five options. (5) Automatic Evaluation: Evaluate open-ended answers in two scenarios— Reference-Free, assessing the correctness of a candidate answer (CA) based only on the visual riddle, and Reference-Based, comparing CAs to the ground truth answer (GTA).

fine-grained image understanding that combines commonsense and world knowledge, making direct comparisons challenging.

**Commonsense reasoning in multimodal models.** Recent progress in vision-language models including models like BLIP2 [8], Flamingo [9], LLaVA [10], GPT4 [11], and Gemini-Pro [12]. These developments have sparked interest in commonsense reasoning [2, 13], leading to complex visual reasoning challenges within the vision-and-language domain. These include specialized tests for understanding associations and analogies [14, 15], interpreting unusual images in WHOOPS! [16], visual abductive reasoning tasks (e.g., Sherlock; [17]), multi-modal humor understanding tasks [3], and world-instructed image-editing tasks requiring commonsense reasoning (e.g., EditWorld; [18]). Notable benchmarks include VisIT-Bench [5] where the authors took existing images and generated challenging questions about them, OK-VQA [19] where questions require world knowledge, and others [20, 21]. However, Visual Riddles distinguishes itself by allowing annotators more creative freedom to devise challenging scenarios and generate corresponding images and questions, rather than restricting them exclusively to what appears in natural images. This also ensures the images have not been previously seen during pretraining.

**Factuality evaluation.** Evaluating factuality has been a focal point in language-only models, particularly for tasks where outputs rely on grounding texts like summarization or machine translation. This domain utilizes reference-based metrics (e.g., Rouge [22], BLEU [23], Bertscore [24], COMET [25]) that compare outputs to a reference, alongside reference-free metrics (e.g., SummaC [26], $Q^2$ [27], TRUE [28], FiC [29]) that evaluate outputs against the input texts. Parallel advancements in vision-language tasks have introduced metrics such as Clipscore [30], AVIBench [31], and Tifa [32], with benchmarks like MileBench [33], SeeTRUE [34], BLINK [35], and Vibe-Eval [36] enhancing the

rigor of factuality assessments. Our dataset, Visual Riddles, extends these challenges into the realm of visual commonsense, requiring both deep world knowledge and nuanced commonsense reasoning to interpret complex visual cues. This requirement marks a significant step beyond traditional benchmarks, like Encyclopedic VQA [6], which predominantly tests factual knowledge against curated text sources. By demanding high sensitivity to visual subtleties alongside robust commonsense analysis, Visual Riddles offers a uniquely stringent test of multimodal model capabilities.

## 3 Data Collection

The Visual Riddles Challenge uses visual clues in images to test common-sense reasoning in vision-and-language models. The goal is to develop a dataset with images that represent ambiguous and culturally rich scenarios. To solve the riddles, models must detect subtle visual clues and engage in reasoning that integrates commonsense and world knowledge. The Visual Riddles benchmark was hand-curated by seven designers, experienced in generating images with text-to-image models. The design process was guided by comprehensive instructions to create visual riddles that were complex enough to challenge models yet solvable by humans. Each riddle consisted of a synthetic image paired with a question and a corresponding ground truth answer (see examples in Fig. 1). To generate high-quality images, the designers had access to advanced text-to-image models, including Midjourney, Ideogram, Canva, Gemini [12], SDXL [37], DALL-E 3 [38], and Stable-Diffusion [39]. The generated images included subtle clues crucial for solving the riddles, and designers were encouraged to embed cultural nuances and their own world knowledge into the riddles. The images were not only photo-realistic but also contained all necessary elements to lead to the correct answer. Designers provided additional hints to the visual clues that guided where to look in the image when trying to solve the riddle (top example in Fig. 1) and, for riddles requiring world knowledge, included attributions to relevant sources (bottom example in Fig. 1). After creation, each riddle was peer-reviewed by at least three other designers to ensure the hint's clarity and the riddle's solvability. Upon approval, the designer drafted a detailed answer that logically explained the solution based on the visual clues. Further elaboration on the image generation process and the instructions for riddle creation are in A.2. The Visual Riddles benchmark is licensed under Apache License 2.0.

**Commonsense and World Knowledge Categorization and Difficulty Analysis**   Finally, we categorize the instances based on the type of commonsense reasoning and knowledge required, including safety-related knowledge, biological knowledge, cultural knowledge, and many more (see A.3). Each riddle is assigned to one of 16 distinct categories, labeled with the single category that best fits it. Additionally, each riddle is assigned a Difficulty Level Index, which quantifies its complexity, ranging from simple, straightforward clues to obscure ones requiring specialized knowledge.

## 4 Visual Riddles Benchmark

This study introduces three vision-and-language tasks within the Visual Riddles benchmark to evaluate model capabilities. The tasks are: solving open-ended visual riddles, choosing the correct answer from multiple options, and conducting automated evaluations on open-ended riddles. These tasks may incorporate auxiliary information such as hints or attributions to enhance assessment accuracy.

**Open-ended VQA**   In this task, models are evaluated on their ability to solve visual riddles comprising of an image and a question, requiring not only correct solutions but also detailed explanations. Questions may be binary, requiring a 'yes' or 'no' answer with justification, or *open-ended*, inviting freely-formulated responses to queries like 'why', 'where', or 'how'. These questions require both locating visual clues in the images and integrating these clues with commonsense reasoning and world knowledge to formulate coherent and correct answers. For example, in Fig. 2.1, a good answer to the question *"What is probably the gender of the cat?"* should reference both the *visual clue* (the cat's fur color) and the *relevant world knowledge* (Calico cats are predominantly female).

**Multiple-choice VQA**   We propose this task as an alternative to the open-ended VQA, which can be evaluated automatically using a simple accuracy metric. It involves selecting the correct answer from five options for a visual riddle comprising an image and a question, as shown in Fig. 2.4. Details on how these candidate answers were collected are provided in §5.3.

**Open-ended VQA Automatic Evaluation** In this task, models are being assessed for their ability to evaluate the accuracy of open-ended responses to visual riddles. As outlined in Fig. 2.5, it has two categories: **(a) Reference-Free**, where the model assesses a candidate answer based only on the image and question; and **(b) Reference-Based**, where it also considers the ground-truth answer to evaluate the candidate's response. This framework aims to identify the best auto-rater for open-ended tasks, supporting automated leaderboard creation. Designed for scalable evaluation of open-ended responses, this task uses vision and language models to judge both Reference-Free and Reference-Based responses. The Reference-Free baseline is included despite available ground truths, as it sometimes outperforms the Reference-Based approach [5], enhancing our insights into model performance across different contexts.

**Auxiliary Information** We utilized auxiliary information within tasks of our benchmark to assess its impact on models performances. Specifically, we incorporate textual hints and source attributions to enhance the model's ability to solve visual riddles by providing targeted contextual cues. Textual hints are concise directives that focus the model's attention on specific visual clues within the image. For instance, a hint such as *"Look at the colors of the fur"* in the calico cat example (see Fig. 2.2) effectively directs the model's focus towards the type of the cat. Additionally, attributions provide essential world knowledge through a webpage URL from which we extract all the text content, offering the models the necessary information to solve the visual riddle. For example, providing a text stating that calico cats are *"almost exclusively female"* (see Fig. 2.3) assists models in more effectively inferring a cat's gender from its fur color.

## 5 Experiments

We evaluate several state-of-the-art publicly accessible vision-and-language models on each of the tasks in Visual Riddles, which we overview in §4. Then, we describe the experimental setup and results of each of the benchmark tasks, including open-ended VQA (§5.2), multiple-choice VQA (§5.3) and open-ended VQA automatic evaluation (§5.4). For riddles evaluated with auxiliary data, which may have lengthy prompts, we only test models capable of accommodating such lengths.

### 5.1 Models

We evaluate LLaVA-1.5-7B [10], LLaVA-1.6-34B [40], InstructBLIP-7B [41], GPT4-turbo-preview [11], Gemini-Pro-Vision [12], and Gemini-Pro-1.5 [7]. We used the most up-to-date versions of each model, as of May 2024. After submission, we included four additional models: GPT4o [42], Claude 3.5 Sonnet [43], Qwen-VL-Max [44], and Molmo-7B [45], all updated to the end of October 2024. Each experiment required prompts of varying token lengths; notably, the prompts for attribution tasks were particularly lengthy, as they also included the attributing texts. Models selection is

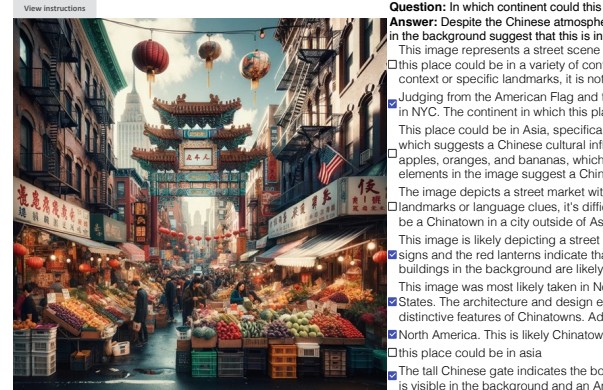

Figure 3: Amazon Mechanical Turk interface for selecting answers to open-ended riddles. Annotators are presented with an image, a question and several candidate answers, including both human-curated and model-generated predictions, and are tasked with identifying the correct responses.

Table 1: Human ratings for large vision and language models, caption generation to large language models pipelines, and human evaluators on 400 riddles in the Visual Riddles benchmark.

| | | % **Human Rating** ↑ |
|---|---|---|
| | Gemini Pro 1.5 | **40** |
| | Gemini Pro Vision | 30 |
| LVLM | GPT4 | 34 |
| | LLaVA-1.6-34B | 15 |
| | LLaVA-1.5-7B | 13 |
| | InstructBlip | 13 |
| Caption→ LLM | Gemini Pro 1.5 → Gemini Pro 1.5 | 23 |
| | Human (Oracle) → Gemini Pro 1.5 | **50** |
| | Humans | **82** |

therefore based on their capacity to accommodate the full prompt required for each task, ensuring that inputs are processed fully without truncation. For further details, see A.4.

## 5.2 Open-ended VQA

We start our investigation by assessing the performance of humans and models on the primary *open-ended* VQA task, which requires solving riddles based solely on the accompanying images. To that end, we collect human responses to the riddles through crowdsourcing, as well as prompting several vision-language models, and evaluate all responses using human judgement.

**Human Answers** We utilized Amazon Mechanical Turk to collect human open-ended answers for the benchmark's riddles. Qualification test ensured ten reliable workers, annotated each of the 400 riddles (three workers per riddle). During annotation, we instructed annotators to not only answer the question but also provide justifications. Annotators were also encouraged to utilize tools like Google Lens and different search engines to research clues and ensure accurate identification of objects. For example, in Fig. 2.4, annotators which are unfamiliar with the Kakapo bird were advised to use Google Lens to identify it. For more details on the annotation process, including full guidelines and UI screenshot examples, see A.5.

**Model Answers** We extract open-ended answers from the models in §5.1 (latest configurations). Two baselines are evaluated: large vision and language models (LVLM) generating answers from image-question prompts, and Caption→ LLM. For the latter, we extract image captions using Gemini-Pro-1.5 and humans (pre-collected), and generate answers for the riddles from caption-question prompts using the best-performing LVLM model. Further details are available in A.4.

**Human Rating of Responses** We assess the correctness of human and model answers using three annotators in a multiple-choice selection format. They select responses that match the ground truth, without any hallucinations, based on provided images, questions, and ground-truth answers. The final rating is determined by a majority vote. Annotator agreement, measured by Krippendorff's alpha [46], reached 79%. An example of human annotation from Visual Riddles is shown in Fig. 3. Further details on annotation guidelines and results can be found in A.6.

**Results** The results, displayed in Table 1, show that models face significant challenges solving the riddles in the Visual Riddles benchmark. In the LVLM category, Gemini-Pro-1.5 is the best performing model with a score of 40%, followed by GPT-4 at 34% and Gemini-Pro-Vision at 30%, while other models perform below 15%. Human performance, by contrast, reaches 82%, underscoring that **these riddles remain a significant challenge for current models**. In the $Caption \rightarrow LLM$ category the best model, Human (Oracle)→Gemini-Pro-1.5, shows a 10% gain over the LVLM, yet it still falls short of human performance. This indicates a recognition gap, with Gemini-Pro-1.5 (23%) providing inadequate captions for images requiring subtle clues, unlike human-generated captions (50%). Further, this shortcoming persists as Gemini-Pro-1.5, even with ground truth captions, improves only to 50%, lagging behind the human rate of 82%. Further results on various categories and difficulty index levels are available in A.3.

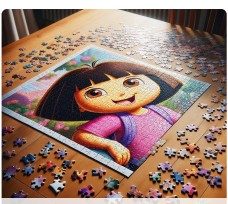

Should I buy this for my toddler?

**Human Caption:** A partially completed **1000-piece** ✗ ✓ jigsaw puzzle ✓ of a **Dora** cartoon spread out on a table.

**Model Caption:** A partially completed **Dora the Explorer jigsaw puzzle** lays on a wooden table, bathed in warm sunlight. The central figure of Dora, with her signature brown bob and a bright smile, is almost complete. She's wearing a pink T-shirt and a purple backpack.
Around Dora, a halo of puzzle pieces forms a border, indicating the image is not yet fully assembled. Numerous loose puzzle pieces are scattered across the table, particularly to the right and top of the image, suggesting an active work in progress. Some pieces are clustered together, hinting at partially formed sections waiting to be connected to the main puzzle. The wooden table provides a warm, textured backdrop, contrasting with the vibrant colors of the puzzle pieces. The sunlight creates a bright and cheerful atmosphere, highlighting the puzzle and the playful nature of the image.

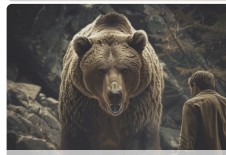

Given that the man doesn't have bear spray, and the bear seems to be aggressive and wanting to attack the man, what should he do?

**Human Caption:** A **man** ✓ **facing** ✓ a **grizzly bear**. ✓

**Model Caption:** The image showcases a powerful **confrontation** between man and nature. A massive **grizzly bear** dominates the foreground, standing tall on a rocky terrain. Its thick, dark brown fur is slightly matted, suggesting a life lived in the wild. Its eyes are fixed forward with a fierce intensity, mouth slightly agape, revealing sharp teeth. This is a creature of immense power, and its presence is imposing. In the background, dwarfed by the bear's size, stands a **man**. He's turned slightly away, but his body language speaks of tension. He wears a simple, earthy brown jacket and appears to be facing the bear with a mix of apprehension and awe. The stark contrast between his vulnerability and the bear's raw power creates a palpable sense of danger and respect. The setting is a rugged, natural environment. Jagged rocks and boulders surround the pair, suggesting a remote and unforgiving wilderness. Muted, earthy tones dominate the image, enhancing the feeling of primal power and the unpredictable nature of a wild encounter.

Figure 4: Comparison of model-generated and human-generated captions that were used in the Caption→LLM setup. 'X' marks captions where critical details are missing in the model-generated version, while 'V' marks captions where these details are present.

**Caption Quality and Model Gaps**  To understand the impact of caption quality on model performance in the Caption→LLM setup, we analyzed 200 riddles (50% of the dataset), comparing human-generated (ground truth) captions with model-generated ones. Human captions, provided by riddle creators to generate the images, typically contained the key elements needed to solve the riddles. For example, Fig. 4 shows a riddle involving a 1000-piece Dora the Explorer jigsaw puzzle. The model-generated caption failed to mention crucial details such as the number of pieces, small size, or complexity, making the question "Should I buy this for my toddler?" unanswerable. Our findings show that 97% of human-generated captions included the necessary information, while only 57% of model-generated captions did. This discrepancy highlights that models often miss important details when captioning, limiting their ability to answer riddles effectively. Even when provided with human-generated Oracle captions, models still struggled with reasoning, achieving only a 50% success rate compared to 23% with model-generated captions, as presented in Table 1. This indicates a 27% "visual recognition gap", where models could improve by better recognizing visual cues, and a 32% "reasoning gap", showing the models' difficulty in reasoning even with perfect visual input. These findings underscore the importance of both accurate captioning and improved reasoning capabilities to close the gap between model and human performance.

**Tool Usage and Model Gaps**  We analyzed all data with attribution, comprising 164 riddles (41%) where external tools, such as Google Search or Lens, could potentially aid in solving. The remaining 59% did not necessitate these tools. Model performance was assessed across both categories, revealing a slight difference: models reached 21% accuracy on riddles where external tools might have been useful and 26% on those not requiring such assistance. These findings suggest that the main difficulties for models lie in reasoning and interpreting visual clues rather than depending on external knowledge.

## 5.3 Multiple-choice VQA

We next evaluate the performance of large vision and language models on the multiple-choice VQA task of the Visual Riddles benchmark, shifting from a generative to a classification task. Each riddle includes an image and a question, along with five answer choices: the correct one from the designer, three incorrect options derived from the human-judgment evaluations of model or human responses to the open-ended riddles (§5.2), and a "cannot determine" option for ambiguous cases. For the 12% instances where fewer incorrect responses were available (i.e., when most responses by humans and models were correct), we used GPT-4 to generate additional distractors, using two in-context examples of visual riddles with incorrect answers. Finally, accuracy is calculated as the proportion of riddles correctly solved by the models, with random guessing yielding a baseline success rate of 20%.

Table 2: Three types of multiple choice evaluations: overall accuracy, accuracy excluding instances where the model selected "cannot determine", and accuracy with auxiliary data (hints or attributions).

| | % Accuracy ↑ | % Cannot Determine | % Accuracy w/o Cannot Determine | + Hint ↑ | + Attribution ↑ |
|---|---|---|---|---|---|
| Gemini Pro 1.5 | 38 | 20 | 48 | 66 | 72 |
| Gemini Pro Vision | 41 | 3 | 42 | 62 | - |
| GPT4 | **45** | 12 | 52 | **69** | **82** |
| LLaVA-1.6-34B | 24 | 8 | 26 | 30 | - |
| LLaVA-1.5-7B | 17 | 0 | 17 | 29 | - |
| Claude 3.5 Sonnet | 46 | 4 | 48 | 45 | - |
| GPT4o | **55** | 17 | 67 | **83** | - |
| Qwen-VL-Max | 35 | 3 | 37 | 53 | - |
| Molmo-7B | 34 | 1 | 35 | 42 | - |

Table 3: Candidate auto-raters performances compared to human rating.

| | Judge | Accuracy of Judge Prediction Compared to Human Rating ↑ |
|---|---|---|
| Reference-Based | Gemini Pro 1.5 | **87** |
| | Gemini Pro Vision | 75 |
| | GPT4 | 86 |
| | LLaVA -1.6-34b | 76 |
| | LLaVA -1.5-7b | 70 |
| Reference-Free | Gemini Pro 1.5 | **52** |
| | Gemini Pro Vision | 38 |
| | GPT4 | 51 |
| | LLaVA -1.6-34b | 43 |
| | LLaVA -1.5-7b | 35 |

**Results** The results in Table 2, indicate that this multiple-choice version is comparably challenging to the open-ended visual riddles, with GPT-4, Gemini-Pro-Vision and Gemini-Pro-1.5 showing the highest accuracies of 45%, 41% and 38%, respectively. Overall, model performance on this task slightly exceeds that on the open-ended task as assessed by human evaluators. Excluding instances where models selected the "cannot determine" option enhances these figures, elevating GPT-4 to 52% and Gemini-Pro-1.5 to 48%. This trend, detailed further in A.7, reveals that models often default to "cannot determine" in the absence of sufficient information. This tendency is mitigated by providing hints and attributions, making GPT-4's accuracy significantly increase to 69% and 82%, respectively. Post-submission results further highlight this trend, with GPT4o leading the models at 55% accuracy overall and reaching 83% with hints.

### 5.4 Open-ended VQA Automatic Evaluation

The automatic evaluation experiments are designed to assess the capability of vision and language models to accurately judge the correctness of open-ended answers to visual riddles. This evaluation is critical for identifying the most effective approach for automated evaluation, supporting scalable future work on our benchmark, and ensuring integration with the leaderboard intended for widespread use by the research community.

**Comparing Auto-Raters to Human Ratings** We evaluate the models in two scenarios: *reference-free* and *reference-based*, in the first scenario, models independently assess the correctness of an answer based solely on the image and its associated question, and in the second, models are additionally provided with the ground-truth answer along with the candidate answer, challenging them to validate the provided answer against the correct one. Each auto-rater candidate was evaluated on a subset of two responses provided by other models and humans, excluding its own responses from this evaluation. This subset consists of balanced responses, including one correct and one incorrect. If one of these options was not available, two responses from the same category were selected. We tested the candidate auto-rater models on the visual riddles annotated in §5.2 to determine which model's performance most closely aligns with human judgments. This alignment is crucial for selecting models suitable for evaluative roles in automated systems.

Table 4: Automatic evaluation of open-ended answers by Gemini-Pro-1.5.

|  | Visual Riddles | | Hints | Attribution |
|---|---|---|---|---|
|  | % Human Rating ↑ | % Auto-Rater Rating ↑ | % Auto-Rater Rating ↑ | % Auto-Rater Rating ↑ |
| Gemini Pro 1.5 | **40** | **53** | **62** | **29** |
| Gemini Pro Vision | 30 | 34 | 47 | - |
| GPT4 | 34 | 38 | 61 | 25 |
| LLaVA -1.6-34b | 15 | 21 | 16 | - |
| LLaVA -1.5-7b | 13 | 19 | 30 | - |
| InstructBlip | 13 | 20 | 28 | - |
| Claude 3.5 Sonnet | - | 39 | - | - |
| GPT4o | - | 50 | - | - |
| Qwen-VL-Max | - | 26 | - | - |
| Molmo-7B | - | 36 | - | - |
| Humans | **82** | **78** | - | - |

**Results**   Table 3 presents the accuracy of the models under two scenarios, with Gemini-Pro-1.5 achieving the highest performance—52% in the reference-free context and 87% in the reference-based context, as measured against human ratings. These results suggest that the reference-free scenario may be less suitable for auto-evaluation, as evidenced by the top score of 52%, indicating a moderate correlation with human judgment. However, the reference-based scenario shows a stronger correlation, with the top two models achieving 87% and 86%. These findings indicate the need for an appropriate judge to assess open-ended answers.

**Utilizing the Optimal Auto-Rater to Evaluate Visual Riddles**   Having established Gemini-Pro-1.5 as the best auto-rater, we utilize it in a reference-based setting to assess all models on the three open-ended settings detailed in §5.2, i.e., the main task, and its hint- and attribution-assisted variants. Table 4 presents the auto-rating accuracy of various models on the open-ended task, with Gemini-Pro-1.5 as the evaluator. Gemini-Pro-1.5 achieves the highest performance, scoring 53% (40% with human rating) on open-ended questions, 62% with hints, and 29% with attributions. These results suggest that models perform better when provided with hints but not with attributions. Hints improve model performance by directing attention to visual clues, but they do not notably enhance reasoning capabilities. Conversely, when attributions are provided, models struggle to filter through irrelevant details, indicating significant challenges in solving open-ended visual riddles compared to human levels, even with auxiliary information. This highlights ongoing difficulties in improving models' visual reasoning capabilities and bridging the gap between human and machine understanding in visual interpretation tasks.

## 6   Analysis

This section explores two key aspects of the Visual Riddles challenge: assessing models' use of visual cues through comparisons between original and modified images, and examining generative models' efficacy in replicating images aligned with our riddle prompts for an automatic generation pipeline.

Table 5: Percentage of correctness rates of ground truth answers for original and altered images. The 'Gap' column quantifies the reduction in performance due to image modifications, illustrating the challenge of context changes for model accuracy.

|  | Original Images↑ | Modified Images↓ | Gap↑ |
|---|---|---|---|
| Gemini Pro 1.5 | 74 | 14 | **60** |
| Gemini Pro Vision | 86 | 51 | 35 |
| GPT4 | 68 | 15 | 53 |
| Llava-1.6-34b | 93 | 53 | 40 |
| Llava-1.5-7b | 54 | 38 | 16 |

### 6.1   Assessing Visual Aspects of Riddles: Ablation Study with Modified Images

To assess models' preference for specific answers, we selected 72 images from our benchmark and created modified versions, altering visual cues to invalidate ground truth answers. This study explores whether models base their answers solely on text or consider visual clues. We conducted two rounds of testing: one with original images and one with modified versions. In both instances, we presented the model with the question, the original ground truth answer, and asked,"Is the answer correct?" (Fig. 5). Post-image modification, known ground truth answers became incorrect, make the evaluation to a binary assessment. For each model, we calculated the percentage of correctness rates on each type of image and the gap. A high gap indicates that the model identified the answer as correct for

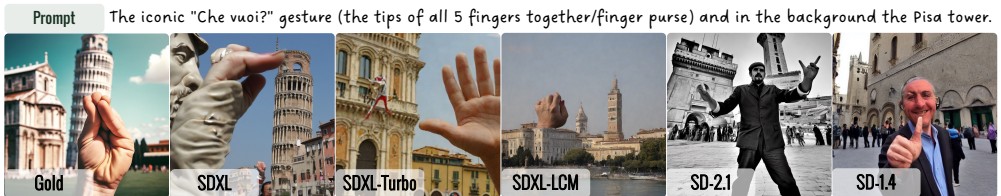

Figure 5: Modified images ablation study: a demonstration of the process where the model evaluates an answer's validity using two scenarios: one with the original image and another with a modified image that alters the visual clue, affecting the correctness of the original ground truth answer.

Figure 6: Reproducing visual riddles: all models fail to capture the nuance in the description.

the original image and incorrect for modified one. Table 5 reveals models' preference for specific answers despite changes in critical image elements and their struggle to identify correct answers, especially with modified images. Gemini-Pro-1.5 excels with a 60% gap, while others below 53%.

## 6.2 The Visual Riddles Prompt Set: A Challenge for Text-to-Image Models

This section shifts our focus from evaluating VLMs and LLMs to showcasing the unique challenge posed by the Visual Riddles prompts to text-to-image models. Designing images for the Visual Riddles dataset, with their specific and often subtle visual clues, proved surprisingly difficult. To quantify this difficulty, we attempted to reproduce 100 visual riddles using five popular open-source diffusion models: SDXL [47], SDXL-Turbo [48], LCM-SDXL [49], SD-1.4, and SD-2.1 [50]. Each model generated 100 images based on these prompts, amounting to 500 images in total. However, only 61 (12%) successfully matched the prompts. The most successful model was SDXL-Turbo, achieving a 15% success rate. A complete breakdown of the models' performance is available in A.8. Fig. 6 showcasing several models struggling to capture the nuances embedded in the prompt. This low overall success rate underscores the unique challenge presented by the Visual Riddles prompts, establishing them as a valuable resource for evaluating and advancing text-to-image generation, especially for tasks that demand high precision and the ability to render subtle visual features.

## 7 Conclusions and Limitations

Our analysis of the Visual Riddles Challenge shows that state-of-the-art vision-language models face difficulties in interpreting complex visual scenarios that require commonsense reasoning. With an average success rate of 40%, models significantly lag behind human performance, which stands at 82%. This performance gap underscores the ongoing challenge of bridging human and machine understanding in complex visual interpretation tasks. Although Visual Riddles is smaller than other benchmarks, it follows a "quality over quantity" approach, seen in recent benchmarks[51, 14, 16, 5, 52, 53], which emphasize high-quality challenges. Our dataset's size allows for meticulous hand-curation to ensure diverse commonsense challenges, focusing on evaluation over training. While we utilized a variety of generative tools to reduce potential biases, this reliance may introduce inherent limitations, and despite efforts to exclude offensive content, some individuals may still find certain riddles inappropriate. Future work could explore automated dataset generation, creating a dedicated training set, and validating Visual Riddles as a robust test set. Generating multiple images for each prompt, coupled with repeated evaluations, would also allow more robust assessments of models' visual reasoning capabilities.

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

# A   Appendix

The subsequent sections offer essential insights into reproducibility, encompassing detailed explanations and examples regarding model evaluation, analysis, and data collection.

For access to the complete Visual Riddles dataset, visit `https://huggingface.co/datasets/visual-riddles/visual-riddles`. Comprehensive information about the dataset fields and loading instructions is provided therein.

## A.1   Reproduciblity and Resources

We executed all evaluation code on a single Colab notebook, accessible on our website `https://visual-riddles.github.io/`. For models using APIs like Gemini and GPT-4, we employed CPU resources. However, for models like Llava and InstructBLIP, which required GPU resources, we utilized a single A100 GPU. We utilized a paid plan for models with APIs, enabling us to generate 400 predictions for each task within a timeframe of less than 30 minutes, equivalent to the runtime of Llava and InstructBLIP models used locally with GPUs. Table 6 lists the models specified in the APIs, and how they were used.

**Auxiliary Data: Attributions**   For each visual riddle containing an attribution field, we extracted the textual content from the webpage URL using Selenium WebDriver (`https://www.selenium.dev/`). Consequently, prompts containing attributions may be lengthy.

Table 6: APIs model specification

| API | Model Name Specified | Type of Model Used | Tasks Used |
|---|---|---|---|
| GPT4 | gpt-4-vision-preview | VLM | Open-Ended, Multiple-Choice (Distractors + Evaluation), Auto-Eval Judge |
| Gemini-Pro-1.5 | gemini-1.5-pro-latest | VLM | Open-Ended, Multiple-Choice, Auto-Eval Judge, Auto-Rater |
| Gemini-Pro-1.5 | gemini-1.5-pro-latest | LLM | Caption $\rightarrow$ LLM on Open-Ended |
| Gemini-Pro-Vision | gemini-pro-vision | VLM | Open-Ended, Multiple-Choice, Auto-Eval Judge |

## A.2   Image Generation Guidelines

The Visual Riddles benchmark was created by seven designers including four women and three men, most of whom are authors of this paper, all from the same country, experienced in generating images using text-to-image models. The annotators were instructed to create images that integrate information with world knowledge and common sense to answer a given textual question. The answer must be grounded in the data presented in the image, making it impossible to respond accurately without comprehending the image and identifying the embedded clues. To generate high-quality images, our designers are given access to advanced text-to-image models, including Midjourney[1], Ideogram[2], Canva[3], DALL-E 3 [38], and Stable-Diffusion [39]. Each image must be sufficiently challenging such that at least one of the evaluation models (e.g., Gemini-Pro-1.5 [7], GPT4-turbo-preview [11] and LLaVA-1.6-34B [40]) fails to correctly answer the question based on the provided image. In addition, the designers provided not only the correct answer to the question but also a hint to the image that should guide where to look in the image when trying to solve the riddle (Example in Fig. 1)

---

[1] `https://www.midjourney.com/home`
[2] `https:///www.ideogram.ai`
[3] `https://www.canva.com`

**Call for AI image designers!**

We are collecting human designers to assist with the generation of a general world knowledge and common-sense challenge data set.

The generation task involves the creation of unique instances each consisting of an image, a question, and an answer. These instances should test the limits of AI's understanding of general world knowledge by presenting scenarios that require multi-step reasoning based on visual cues. The main goal is to craft questions that an AI model cannot easily answer just by analyzing the image and the text, thereby pushing the boundaries of current AI capabilities.

The images should be in a **.jpg** format 1024x1024 size, the textual file should contain the question, the answer, the relevant image file name for each instance, and the name of the models that you were used to generate each of your images. This file should be in a **.csv** format:

| Question | Answer | Image file name | Image caption | Generative Model name | Hint | Attribution |
|----------|--------|-----------------|---------------|-----------------------|------|-------------|

**Examples for generative models:**
https://ideogram.ai/t/top/1, https://screenshot.googleplex.com/333oLFUd2FuhfPZ ,
https://stablediffusionweb.com/#google_vignette , https://www.canva.com/ai-image-generator/ ,
https://www.midjourney.com/home , https://gemini.google.com/app

**Image Creation**
*Realism and AI Generation:* Each image should be crafted to look as realistic as possible, utilizing advanced AI-based image generation tools such as Dall-E, Midjourney, or Stable Diffusion etc. The chosen imagery should directly relate to the question and answer, serving as a visual foundation for the challenge.
*Accuracy Verification:* After generating an image, ensure that it accurately represents the intended scenario or concept (the concept must be a real-world one, don't use ideas that do not happen in reality). The image should visually encapsulate all the elements necessary to lead to the question and, by extension, the answer.

**Crafting Questions**
*Challenge Through Indirect Connection:* Questions should be designed to challenge AI models by requiring two-step logical reasoning (two-hop) based on the information depicted in the image. For instance, an image showing leafless trees and lush greenery suggests a winter scene, which implies colder temperatures, leading to questions about appropriate attire for the conditions.
*Specificity and Relevance:* Each question must directly relate to the visual and conceptual content of the image, prompting the responder to make logical connections between what is seen and the broader implications or facts of the world.
*additional note* : verify that each question cannot be answered without the relevant image.

**Formulating Answers**
The answers can be short and free-style, please follow these:
*Detail-Oriented Responses:* Answers should be comprehensive, detailing all aspects of the image relevant to the question. This includes describing the visible elements that lead to the understanding necessary to address the question accurately.
*Completeness and Advisory:* Beyond just stating facts, answers should also provide a conclusive statement or recommendation based on the question posed. The response should leave no ambiguity about the appropriate course of action or the logical conclusion to be drawn from the image-question context.
**PAY ATTENTION: the answer should refer to both the visual clue in the image, and the commonsense/world knowledge that is required for the solution.**

**Hint:** a textual hint that directs the solver (human or model) for the visual clue. cannot be the clue itself.

**Attribution:** for questions that their answer requires a "reliable source" like answers that talk about rules/countries/biological conditions/cultural rules/etc, there is a need to provide a link for a websource that confirms your answer (does not have to be only wikipedia).

Figure 7: Guidelines for human designers to create a visual riddle — Part 1. This section includes instructions on the visual riddle creation process including requirements for captions, generative models, hints, and attribution.

and, for riddles requiring world knowledge, include attributions to relevant sources (see the full instructions in Fig. 7, Fig. 8 and the bottom example in Fig. 1). After creation, each image-question pair undergoes a peer review by at least three other designers to ensure the hint's clarity and the riddle's solvability.

## A.3 Categories and Difficulty Level: Breakdown and Results

To examine how different types of world knowledge and difficulty levels impact model performance, we analyze their correlations to identify particularly challenging combinations for current models. The distribution of the different categories and difficulty level (with 3 - most difficult and 0 - least

Examples:

1. **Q: He wants to go outside, what should he wear?**
   **A**: "In his window, there are leafless trees and high greenery, features characteristic of the winter season. Therefore, despite the sunny appearance, the temperature could be low. It is advisable to wear warm clothing when going outside."
   **H**: Look at the plants outside.
   **At**: https://forestryandland.gov.scot/blog/trees-in-autumn

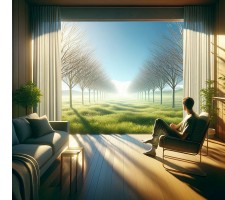

2. **Q: What is the language she would probably speak in this place?**
   **A**: Salmon and cream cheese bagel is a popular food in NY, USA. so she probably will speak english there.
   **H**: What does she eat?.
   **At**: https://www.alphafoodie.com/the-best-lox-bagel/

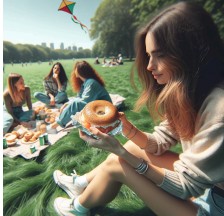

3. **Q: Why does she look like this?**
   **A:** The girl in the image holds a balloon, and her hair is standing up. When a balloon is rubbed against hair, electrons are transferred from the hair to the balloon, creating a static charge that causes the static hair phenomenon. So the reason for the appearance of standing hair is static electricity.
   **H**: what is she holding?
   **At**: https://en.wikipedia.org/wiki/Static_electricity

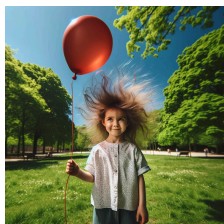

4. **Q: How to get out?**
   **A**: There is a bat in this cave, therefore a way out of the cave is the way that the bat came in through. So, to get out, one must follow the entrance-path of the bat.
   **H**: What can you see in the cave?

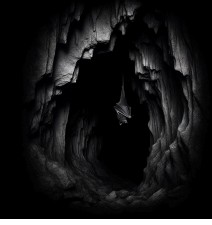

These examples illustrate the intended structure and depth of each instance within the dataset. Designers are encouraged to use this as a template for creativity, ensuring that each element of the instance—the image, question, and answer—works harmoniously to challenge and enhance AI understanding of complex, real-world knowledge.

Figure 8: Guidelines for human designers to create a visual riddle — Part 2. This section includes examples, and, similarly to the presented examples, designers were asked to provide additional details such as the image caption, the generative model used, a hint, and attribution for their riddles.

difficult) are illustrated in Tables 7 and 8. Fig. 9 presents a heatmap illustrating model failures and human failures across various categories and difficulty levels within the Visual Riddles challenge. Across 16 categories and 4 difficulty levels, our analysis identifies model weaknesses in categories like "Object Counting", "Temporal Principles", and "Physical Principles". Notably, human performance on "physical principles" appears to be even more susceptible to errors compared to the models on difficulty index level of 3. Naturally, models as well as humans also seem to struggle more with the higher difficulty instances (levels 2 and 3 in Fig. 9), highlighting the most demanding elements of the benchmark for current vision-and-language models. Yet this highlight the gaps between humans and models performances.

To provide a clearer understanding of the performance across difficulty levels, we included aggregated results for open-ended VQA accuracy of both humans and models, categorized by different categories and difficulty index levels (see Figure 10). The analysis shows that certain categories, such as Temporal Principles, Object Counting, and Biological Principles, pose greater challenges for models, while Geographic Knowledge and Physical Principles are more difficult for humans. Interestingly,

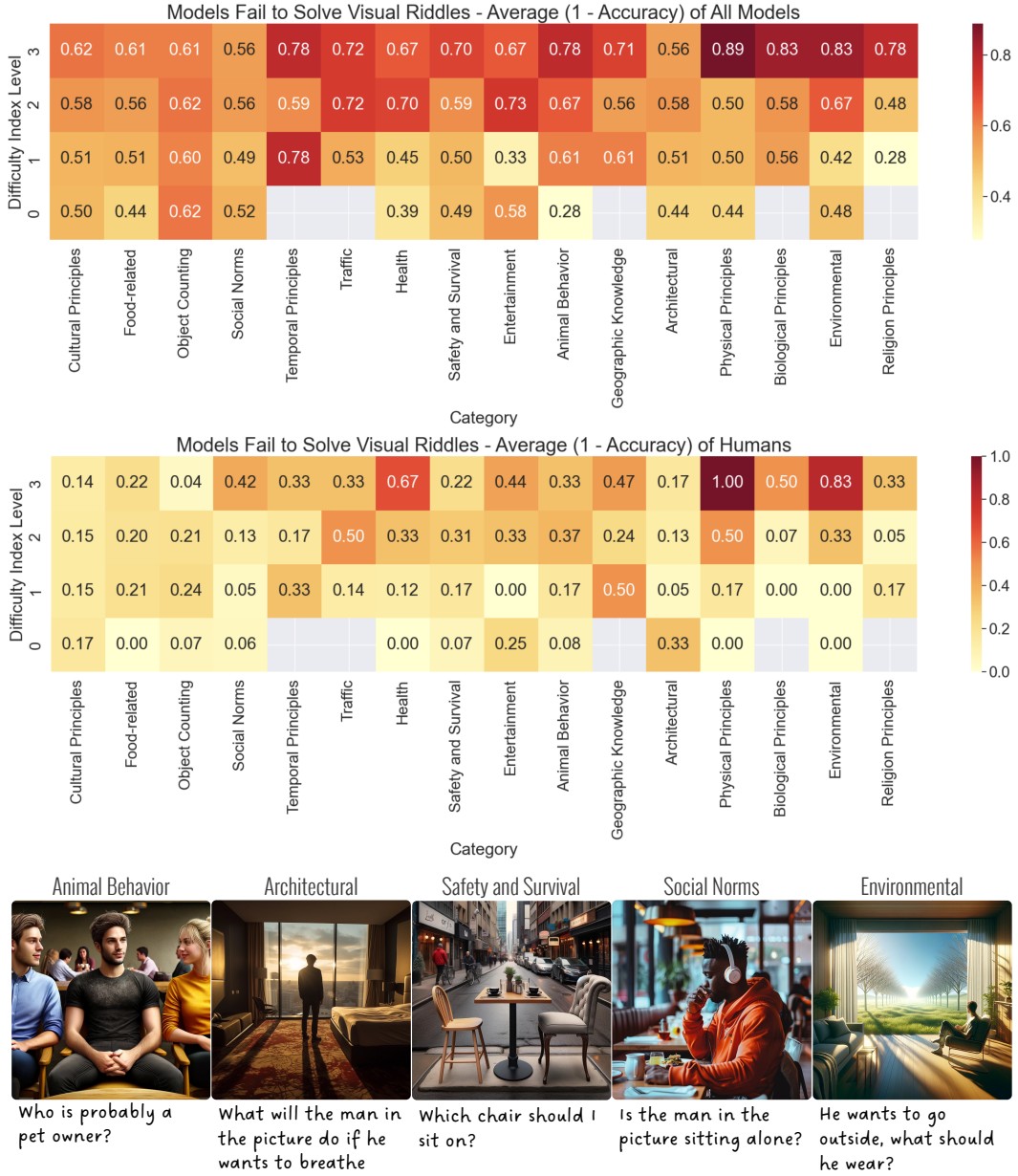

Figure 9: A heatmap depicting how models (Top) and human annotators (Middle) fail to solve visual riddles in the Visual Riddles dataset, visualizing by the color intensity of each cell. The x-axis represents different categories of riddles, while the y-axis shows the difficulty index levels. Red cells highlight particularly challenging areas, with vertical red bands indicating categories that consistently pose difficulties and horizontal red bands confirming the appropriateness of the difficulty index levels assigned. This visualization underscores the diverse range of commonsense knowledge needed to effectively tackle the visual-riddles. Below, there is an example of images related to some of our categories.

there is no significant correlation between the difficulty categories for humans and models, with a correlation coefficient of -0.12. For difficulty levels, while human performance consistently surpasses models, both show a decline as difficulty increases, with a strong correlation of 0.94 (see top and middle parts of Fig. 9). To further illustrate, examples of images corresponding to different difficulty levels are provided in Fig. 10 (bottom).

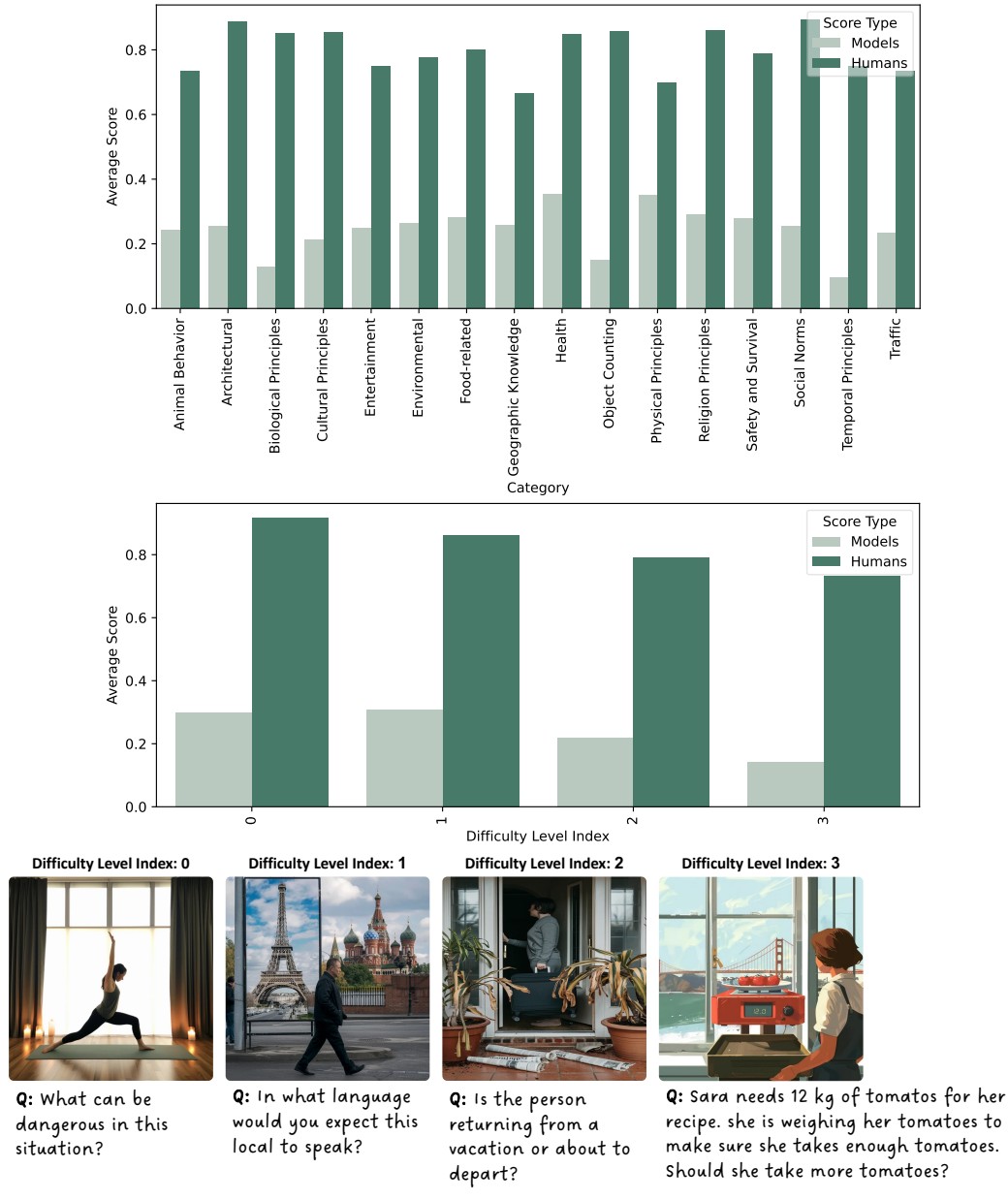

Figure 10: Aggregated results for model and human performance across categories (top) and difficulty levels (middle), along with visual riddle examples illustrating varying difficulty levels (bottom). The top section shows the average open-ended VQA accuracy for models and humans by category, with a correlation of -0.12 between their performances. The middle section presents the average open-ended VQA accuracy for models and humans by difficulty level, showing a correlation of 0.94.

## A.4 Prompts for Different Models

There are several tasks with different prompts:

**Multiple-Choice VQA:** A specific prompt is used for generating distractors when there are insufficient incorrect answers using GPT-4. Additionally, for multiple-choice VQA, we evaluate the models' performance across three settings: (1) given only the image, the question, and five possible answers (with only one correct answer), (2) given the image, the question, the possible answers, and a hint,

Table 7: Distribution of Categories Across Images

| Category | % Images |
|---|---|
| Cultural Principles | 19.50 |
| Social Norms | 15.75 |
| Safety and Survival | 8.25 |
| Food-related | 7.50 |
| Object Counting | 7.00 |
| Health | 6.00 |
| Traffic | 5.00 |
| Animal Behavior | 5.00 |
| Geographic Knowledge | 4.50 |
| Entertainment | 4.00 |
| Architectural | 3.75 |
| Temporal Principles | 3.00 |
| Environmental | 3.00 |
| Religion Principles | 3.00 |
| Physical Principles | 2.50 |
| Biological Principles | 2.25 |

Table 8: Distribution of Categories Across Images

| Difficulty Level | % Images |
|---|---|
| 0 | 11.10 |
| 1 | 33.25 |
| 2 | 38.25 |
| 3 | 17.50 |

and (3) given the image, the question, the possible answers, and an attribution. The prompts structure is outlined in Table 9.

**VQA Automatic Evaluation:** To find the best judge (auto-rater), we evaluate the models in two scenarios - *reference-free* and *reference-based*. The prompts structure is outlined in Table 10.

**Auto-Rater for Open-Ended VQA and Ablation Study with Modified Images:** Using the best Auto-Rater we evaluate all models automatic ratings on the Open-Ended VQA task. We also use the same prompt in our analysis we perform ablation study to explores whether models base their answers solely on text or consider visual clues. The prompts structure is outlined in Table 10.

### A.5 Open-Ended VQA: Annotators Solve Visual Riddles Guidelines

In order to evaluate how well humans are capable in solving the visual riddles questions, a human response were collected using Amazon Mechanical Turk platform www.mturk.com. To gather human responses for the benchmark's riddles, we used the Amazon Mechanical Turk platform, paying annotators $18 per hour. We contacted workers with a proven track record in similar tasks and invited them to a qualification round that began with a review of task guidelines and included solving five riddles of varying difficulty. Following an assessment of their responses and providing personalized feedback, only those demonstrating a strong understanding of the task qualified. Of 14 candidates, 10 proceeded to the actual annotation, where each of the 400 riddles was solved by three annotators. In the guidelines, annotators were presented with five examples. For each example, they were initially shown the visual riddle and subsequently given the solution and the process for solving the question. We recommended that annotators first attempt to solve the riddles on their own before reviewing our provided answers.

Two examples of the guidelines are in Fig. 11 and Fig. 12. In the first, answering the question required only common-sense and counting capabilities of objects in the image while in the second example, world knowledge about cultural principles is needed therefore, the workers were expected to search the different items, understand the clues given in the image and only then answer the question.

Table 9: Prompts for different models for Multiple choice VQA

| Task | Prompt |
|---|---|
| Creating Distractor | "Here is a question regarding the image, and a ground-truth answer. \n Question:$< question >$ \n Ground-Truth Answer $< ground\_truth\_answer >$ \n\n Please generate $< num\_of\_incorrect\_distractors >$ wrong answers (that are kind-of similar to the ground-truth answer, and in a similar length) to the question based on the image, in the format of: \n\n YOUR FIRST ANSWER@@@YOUR SECOND ANSWER@@@..." |
| Clean | "This is a multiple-choice question concerning the image. Out of the options labeled (1)-(5), only one is correct. Please provide your answer as a single digit that corresponds to the correct option. For instance, if the correct answer is (3), you should respond with 3. \n\n Question: $< question >$ \n\n Candidate answers: \n (1)$< candidate1 >$ \n (2)$< candidate2 >$ ..." |
| + Hint | "This is a multiple-choice question concerning the image. Out of the options labeled (1)-(5), only one is correct. Please provide your answer as a single digit that corresponds to the correct option. For instance, if the correct answer is (3), you should respond with 3. \n\n Question: $< question >$ \n\n Hint: $< Hint >$ \n\n Candidate answers: \n (1)$< candidate1 >$\n (2)$< candidate2 >$ ..." |
| + Attribution | "This is a multiple-choice question concerning the image. Out of the options labeled (1)-(5), only one is correct. Please provide your answer as a single digit that corresponds to the correct option. For instance, if the correct answer is (3), you should respond with 3. Additionally, an attribution, which is a textual content from a webpage providing the basis for the correct answer, is also included below. Use this information to select the correct answer. \n\n Question: $< question >$ \n\n Attribution: \n\'\'\'\n$< attribution >$\n\'\'\'\n\n\n Candidate answers: \n (1)$< candidate1 >$\n (2)$< candidate2 >$ ..." |

Table 10: Prompts for different models for Automatic Evaluation

| Task | Prompt |
|---|---|
| Reference-Free, Ablation Study with Modified Images | "Answer with only Yes OR No. Given the image and the question, is the candidate answer correct? \n Question: $< question >$ \n Candidate Answer: $< candidate\_answer >$ \n" |
| Reference-Based, Auto-Rater for Open-Ended VQA | "Answer with only Yes OR No. Given the image, the question and the ground-truth answer, is the candidate answer correct? \n Question: $< question >$ \n Ground-Truth Answer:$< ground\_truth\_answer >$ \n Candidate Answer: $< candidate\_answer >$ \n" |

In Fig. 13 there is an example of the UI page for annotation of solving a visual riddle.

## A.6 Open-Ended VQA: Humans and Models Answers Annotation Evaluation Guidelines

In order to evaluate how well models and humans answered the visual riddle, we used MTurk platform, paying 18$ per hour. We contacted workers and invited them to a qualification round that began with a review of task guidelines and included solving five riddles of varying difficulty. Following an assessment of their responses and providing personalized feedback, only those with a strong understanding of the task moved on to the actual annotation phase. Of 9 candidates, 6 proceeded to the actual annotation phase. To ensure greater cultural diversity in the benchmark, all AMT annotators were selected from countries different from those of the visual riddles creators.

### 1. Visual Riddle

**Question**: How many bracelets are there in the picture?

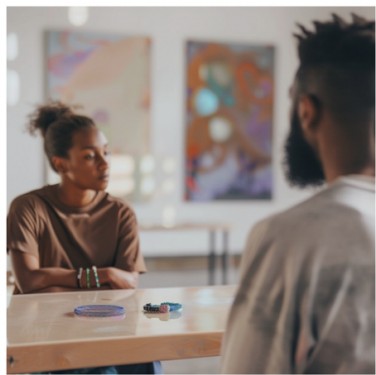

### 2. Answer

**Answer**:
there are two bracelets on the table. in addition, the person in the back is wearing three more bracelets. therefore, there are in total 5 bracelets in the picture.

### 3. Explanation

**To correctly answer many such riddles, you would have to observe the entire image, and not only what is in the front. Many of the hints would appear in the background or any other location that is not in plain sight, and it'd be you job to pick up on all those hidden hints.**

Figure 11: Example of the guidelines for MTurk annotators for solving a visual riddle as an open ended question. (1) Question: Solving visual riddles composed of images and open-ended questions. (2) Answer: the answer to the question. (3) Explanation & Notes: explanation of why this is the correct answer given the question and the image.

During annotation, In the LVLM and the Caption → LLM cases, each annotator was provided with the image, the question, and the correct answer, along with the candidate answers from six different models and three humans (the human answers were obtained as described in §5.2). In the same way (image, question and ground truth answer), we annotated the candidate answers for human (oracle) and Gemini-Pro-1.5 captions from the best model as described in §5.2.

In the guidelines, annotators were presented with five examples. For each example, they were initially shown the image, the question, the gold-answer, and the candidate answers. Afterwards, they could see which answers were correct and the reasoning behind the correctness or incorrectness of each candidate answer. Marking a candidate answer as correct required that it not only accurately answered the question based on the given answer but also contained no hallucinations.

Figure 12: Example of the guidelines for MTurk annotators for solving a visual riddle as an open ended question. (1) Question: Solving visual riddles composed of images and open-ended questions. (2) Answer: the answer to the question. (3) Explanation & Notes: explanation of why this is the correct answer given the question and the image. (4) Attribution: for this image a world knowledge is required therefore - a search in google is helpful in getting the data.

An example of the guidelines as well as a UI page is in Fig 14. Using these annotations as mentioned in §5.3, we compose the Multiple-Choice VQA prompts by sampling three answers annotated as incorrect. If there are fewer than three, we generate additional incorrect answers. We then include the ground-truth answer provided by the riddle designer, along with the "cannot determine" option.

**Instructions:** Given an image and a question, answer the question based on the image.

**Important:** Make sure you look for hints and details that are hidden within the image.

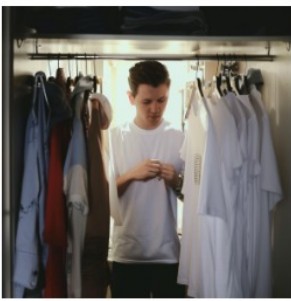

**Question: Does Dani have any black clothes?**

Answer the question based on the image

_______________________________________________

[ Submit ]

Figure 13: Example of MTurk UI page for answering the question given an image and a question

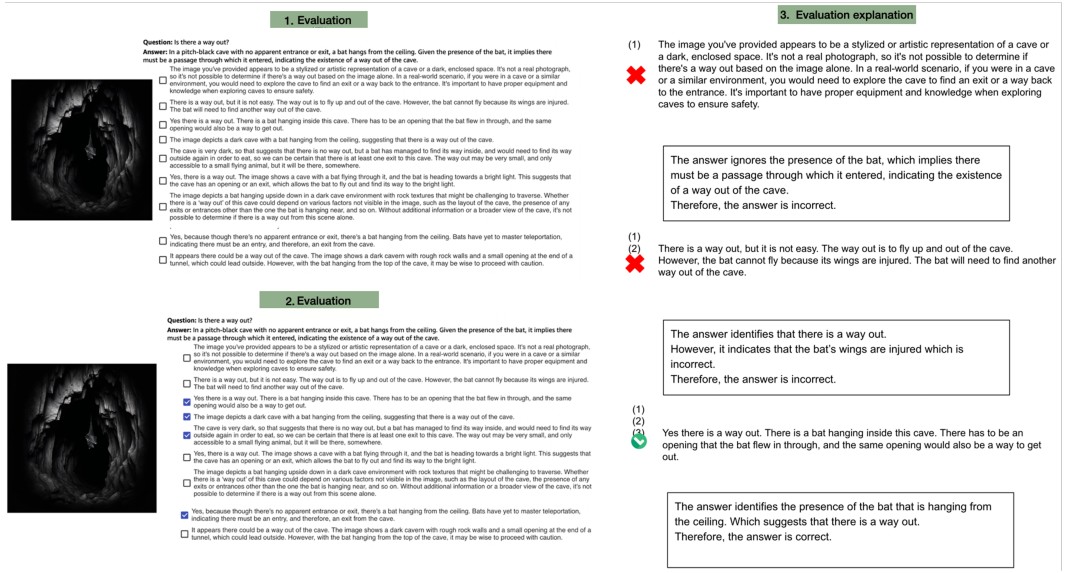

Figure 14: Example of guidelines for MTurk annotators evaluating different answers to the visual riddle. (1) Annotator UI page displaying the visual riddle, the correct answer, and the candidate answers. (2) Solution: Identification of which candidate answers correctly solve the visual riddle. (3) Explanation & Notes: Detailed reasoning for the correctness or incorrectness of each candidate answer (only explanations for three candidates are shown due to space constraints).

## A.7 Multiple Choice VQA Analysis: Model Hesitation

As mentioned in §5.3, a notable occurrence was the frequent selection of the "cannot determine" option by some models. Further analysis excluding these instances revealed improvements in comparison to the results with this option (see Table 2), with GPT-4 achieving 52% accuracy instead of 45%, and Gemini-Pro-1.5 reaching 48% accuracy instead of 38%, respectively. These results suggest that **some models hesitate to answer certain questions, opting for the "cannot determine" option when lacking sufficient information to decide.** Additionally, as shown in Table 11, providing models with auxiliary information via hints and attributions reduces their likelihood of selecting the "cannot

determine" option. For example, Gemini-Pro-1.5 achieves an accuracy rate of 74% with a hint, while GPT-4 achieves an accuracy rate of 84% with an attribution.

Table 11: Percentage of "Cannot Determine" Answers

| | % Cannot Determine | + Hint % Cannot Determine | + Attribution % Cannot Determine |
|---|---|---|---|
| Gemini Pro 1.5 | **20** | **12** | **9** |
| Gemini-Pro-Vision | 3 | 2 | |
| GPT4 | 12 | 3 | 3 |
| LLaVA-1.6-34B | 8 | 10 | |
| LLaVA-1.5-7B | 0 | 0 | |
| Claude 3.5 Sonnet | 4 | 1 | |
| GPT4o | 17 | 4 | |
| Qwen-VL-Max | 3 | 1 | |
| Molmo-7B | 1 | 1 | |

Table 12: Accuracies Excluding "Cannot Determine" Answer

| | % Accuracy w/o Cannot Determine | + Hint % Accuracy w/o Cannot Determine | + Attribution % Accuracy w/o Cannot Determine |
|---|---|---|---|
| Gemini Pro 1.5 | 48 | **74** | 79 |
| Gemini-Pro-Vision | 42 | 64 | |
| GPT4 | **52** | 71 | **84** |
| LLaVA-1.6-34B | 26 | 34 | |
| LLaVA-1.5-7B | 17 | 29 | |
| Claude 3.5 Sonnet | 48 | 45 | |
| GPT4o | **67** | **87** | |
| Qwen-VL-Max | 37 | 53 | |
| Molmo-7B | 35 | 43 | |

## A.8 The Visual Riddles Prompt Set: A Challenge for Text-to-Image Models

n this section, we present the complete breakdown of the models' performance in generating images that fit the visual-riddles prompts. Table 13 presents the full results, indicating that models struggle to generate images that include delicate hints hidden within them, with the best-generating model, SDXL-Turbo [48], creating only 15% of images that match the given prompt.

Table 13: Model success rates

| | % Generation Success ↑ |
|---|---|
| SD-1.4 | 7 |
| SD-2.1 | 12 |
| SDXL-LCM | 12 |
| SDXL | 14 |
| SDXL-Turbo | **15** |

## B Discussion any potential negative societal impacts Visual Riddles

While Visual Riddles offers a unique platform for enhancing visual reasoning and commonsense understanding, it also presents potential challenges that merit careful consideration. One concern is the inadvertent reinforcement of biases. Despite rigorous efforts to design inclusive and culturally neutral visual riddles, the possibility remains that some content might unintentionally reflect or amplify societal stereotypes. Moreover, the complexity of certain riddles could disadvantage users with specific cognitive or sensory impairments, thereby limiting their participation and representation in the research facilitated by this benchmark.

Another limitation is the reliance on automated evaluation methods. Such methods may not fully capture the depth of human reasoning and interpretation, potentially affecting the robustness and transparency of the evaluations. This might inadvertently prioritize certain types of reasoning over others, skewing the development and assessment of AI systems.

To address these issues, we are committed to a continuous review process involving diverse stakeholders to help identify and mitigate any biases or exclusions. This process will include refining the riddles and improving the evaluation methods to ensure they are as inclusive and representative as possible. Additionally, ongoing adjustments will be made to the evaluation protocols to enhance their ability to assess nuanced and complex responses, thereby ensuring that the benchmark remains a fair and effective tool for advancing AI research in an ethically responsible manner.

## C   Designer Consent

We acknowledge and extend our gratitude to all designers who contributed to the benchmark. The credit for each visual riddle is included as part of our dataset. All designers have consented to contribute their creations to this research.

