# OpenReview forum: "Visual Riddles: a Commonsense and World Knowledge Challenge for Large Vision and Language Models"
_NeurIPS.cc/2024/Datasets_and_Benchmarks_Track — NeurIPS 2024 Track Datasets and Benchmarks Poster_

### Official Review · Reviewer_5gUA · 2024-07-23

**Rating:** 7
**Confidence:** 4
**Correctness:** No concerns
**Clarity:** The paper is well written and easy ti…

**Review:**

The paper contributes to better evaluation of vision-language models on more challenging tasks, as well as common-sense reasoning and world knowledge. The paper is well-written and easy to follow.

Strengths:
- The data collection and benchmark design is generally well thought out.
- Giving the riddle creators access to multiple tools and asking them to embed their own cultural knowledge can ensure the benchmark is not biased towards one t2i model or culture.
- Multiple open-sourced VLMs are evaluated
- Multiple tasks are considered, the the use of VLMs for autoevals is investigated and evaluated.

Weaknesses:
- To ensure the benchmarks is actually diverse, it would be good to see a breakdown of 1) which generative image tools were used by creators, and 2) the background of creators, e.g. by continent/ethnicity/race, etc.
- Table 1:  % human rating for the column name is a bit confusing. Humans have a 82% human rating – does that mean 82% accuracy according to the GT?
- Difficulty levels: The difficulty levels are a bit unclear to me and the results are not super conclusive. What is the aggregated performance across difficulties for humans and models (‘average’ of Figure 6 in the appendix). Can you show examples of images of different difficulties?
- More examples. Further to the point above, it would be good to see more examples of the visual riddles in the paper or website, together with the correct answers. For some of the ones shown in the appendix, I myself am not sure what the correct response is.
- Given the very small dataset size, it would be good to provide some sort of statistical analysis on the results / elo ratings / aggregate evaluations over multiple seeds

**Strengths:**

See the review. I think that the paper provides a good contribution to the community, where performance on many benchmarks is becoming saturated and models make minimal gains.

**Additional Feedback:**

Look at weaknesses

**Documentation:**

Yes

**Ethics:**

No ethical concerns

**Limitations:**

Limitations look good. My only concern is the mention of diversity of the tools used and the annotators/designers involved, which is not quantified.

**Opportunities For Improvement:**

See weaknesses

**Relation To Prior Work:**

Prior work has been sufficiently discussed.

**Summary And Contributions:**

This paper introduces a dataset that tests VLM’s performance on VQA that requires spotting subtle visual cues to give the right answer. The benchmarks consists of 400 “visual riddles”, where images have been created by a t2i model. The paper finds that the performance of frontier models falls behind human performance.

---

> ### Author Rebuttal · Authors · 2024-08-16
>
> > *To ensure the benchmarks is actually diverse, it would be good to see a breakdown of 1) which generative image tools were used by creators, and 2) the background of creators, e.g. by continent/ethnicity/race, etc.*
>
> We used various generative image tools, including Midjourney, Ideogram, Canva, Gemini, SDXL, DALL-E 3, and Stable-Diffusion, detailed in Section 2 ("Data Collection") and Appendix A.2 ("Image Generation Guidelines"). Each image's creation tool is listed alongside the image in our dataset and explorer, and this will be further highlighted in the camera-ready version of the paper.
> Our creators team consists of four women and three men, all from the same country and with basic experience in text-to-image models, with some having a computer science background. This information, along with the creators' comprehensive backgrounds, will be added in the camera-ready version. To ensure diverse perspectives, we selected annotators from different countries than the creators, primarily from the USA, South America, and the UK, to enhance the benchmark's diversity across cultural contexts.
>
>
>
> > *Table 1: % human rating for the column name is a bit confusing. Humans have a 82% human rating – does that mean 82% accuracy according to the GT?*
>
> Yes, the 82% human rating refers to the accuracy as evaluated by humans, contrasting with the automated ratings discussed in Table 4. We provide a detailed explanation of the human rating metric in Section 4.2 under "Human Rating of Responses."
>
>
>
> > *Difficulty levels: The difficulty levels are a bit unclear to me and the results are not super conclusive. What is the aggregated performance across difficulties for humans and models (‘average’ of Figure 6 in the appendix). Can you show examples of images of different difficulties?*
>
> Thank you for your feedback! In response, we have included aggregated results for the open-ended VQA accuracy of both humans and models, categorized by content categories and difficulty index levels, in the attached PDF file (Figure 1), which will also be included in the camera-ready version. The analysis shows that categories like Temporal Principles, Object Counting, and Biological Principles are more challenging for models, while Geographic Knowledge and Physical Principles are more difficult for humans, with no correlation between the two (-0.12). For difficulty levels, human performance consistently surpasses models, though both decline as difficulty increases (correlation of 0.94). To clarify further, we include examples of images for different difficulty levels in the attached PDF (Figure 2) and in the appendix of the revised paper.
>
> >*More examples. Further to the point above, it would be good to see more examples of the visual riddles in the paper or website, together with the correct answers. For some of the ones shown in the appendix, I myself am not sure what the correct response is.*
>
> Thank you for your suggestion to include more examples of visual riddles. The full dataset was attached in the supplementary materials. We added additional examples in the attached PDF file. All riddles, along with their corresponding information and correct answers, are fully attached in the supplementary materials, and available in the project website (dataset and explorer), and the links will be available in the camera ready version.

---

> > ### Author Response · Authors · 2024-08-23
> >
> > Dear Reviewer 5gUA,
> >
> > Thank you again for your constructive feedback. As the discussion period draws to a close in one week, we sincerely welcome any further feedback you may have. We deeply appreciate the time and effort you have already dedicated to reviewing our work.
> >
> > Sincerely,
> >
> > The Authors

---

> > > ### Comment · Reviewer_5gUA · 2024-08-27
> > > **Thanks for the response**
> > >
> > > Thank you for the response; I will keep my original rating (7, accept)

---

### Official Review · Reviewer_zua6 · 2024-07-27
**New data set for evaluating common sense and world knowledge in LVLMs**

**Rating:** 9
**Confidence:** 3
**Correctness:** As far as I can judge, the claims mad…
**Clarity:** The paper is written very well.

**Review:**

The paper is of high quality. Its clarity and originality make a significant contribution to the track. Even if one can argue that visual riddles are not directly related to real-world applications, implications become apparent when the categorization of common sense and world knowledge are discussed.

**Strengths:**

The authors nicely explain the data set creation and curation process. They also discuss various evaluation possibilities and their pros and cons. They give analysis results from (only) 5 models but analyze them in detail. The data set is available online for further research.

**Additional Feedback:**

None.

**Documentation:**

Yes, there is a sufficient documentation of everything.

**Ethics:**

The authors mention potential ethical implications.

**Limitations:**

The limitations are considered.

**Opportunities For Improvement:**

In Table 5, there are inconsistent numbers for Llava-1.6-34b.

**Relation To Prior Work:**

The relation to prior work could be discussed earlier in the paper to allow for a more informed discussion on how the data set contributes and exceeds the state of the art.

**Summary And Contributions:**

The paper introduces a new data set (newly generated images) to evaluate common sense and world knowledge reasoning and incorporates small visual clues for LVLMs. With the images, text data like hints, attribution, and ground truth answers are created to provide a whole set of different evaluation options. They also provide a kind of test suite with automatic evaluation, either reference-free or reference-based.

---

> ### Author Rebuttal · Authors · 2024-08-16
>
> Thank you for your thoughtful review and for highlighting the strengths of our paper. We appreciate your positive comments on the quality, clarity, and originality of the work, as well as your recognition of the dataset creation and detailed analysis. Your mention of the various evaluation possibilities and the availability of the dataset for further research is particularly encouraging. We're also grateful for your appreciation of how the paper contributes to the discussion of common sense and world knowledge. We will work to address the points you raised to further improve the paper.
>
>
> > *In Table 5, there are inconsistent numbers for Llava-1.6-34b*
>
> Thank you for pointing out the inconsistency in Table 5 regarding the Llava-1.6-34b data. You are correct; the issue stemmed from a typographical error—a '2' was mistakenly noted instead of a '4'. We will correct this in the camera-ready version of the paper.
>
>
> > *The relation to prior work could be discussed earlier in the paper to allow for a more informed discussion on how the data set contributes and exceeds the state of the art.*
>
> This is a great point that we will improve. Visual Riddles focuses on fine-grained image understanding and commonsense reasoning, setting it apart from other VQA datasets like WHOOPS!, Sherlock, VisIT-Bench, and Encyclopedic-VQA. We discuss this in Section 6, and we will ensure this discussion appears earlier in the camera-ready version.

---

### Official Review · Reviewer_g9pM · 2024-07-30
**Ingenious benchmark Visual Riddles for Large Vision and Language Models**

**Rating:** 7
**Confidence:** 3
**Correctness:** Correct.
**Clarity:** Well-written paper.

**Review:**

The specific contents are described in detail below.

**Strengths:**

Design a novel benchmark for large vision and language models.
Evaluate the performance between large models and humans.
Experienced designers are involved in the construction of the dataset.

**Additional Feedback:**

N/A

**Documentation:**

The authors describe enough details to support benchmark reproducibility. And a beautiful URL for reader access to the dataset.

**Ethics:**

No need.

**Limitations:**

Limitations and potential negative societal impact are discussed, and the authors hope to address these issues in the future. Detailed discussion is provided in Opportunities For Improvement

**Opportunities For Improvement:**

The authors should consider adding a comparison of the differences between this work and other visual question answering datasets.

I think it's great that experienced designers are involved in building the dataset, but the size of the dataset still seems too small. It is also worth discussing whether the fine-tuning of the existing open source model can help improve the effect of the multi-modal large model in this benchmark.

Figure 2.5: What is the meaning of CA and GTA?

**Relation To Prior Work:**

Clear discussion.

**Summary And Contributions:**

This work carefully designed a new dataset and QA benchmark called Visual Riddles. The data for the Visual Riddles benchmark is generated from the text-to-image model, including Midjourney, ldeogram, Canva, Gemini, SDXL, DALL-E 3, and Stable-Diffusion. In particular, seven designers experienced in image generation were deeply involved in building the dataset. Experimental evaluation involved six multimodal large-scale models and human volunteers, and the authors provide a detailed analysis of the results.

---

> ### Author Rebuttal · Authors · 2024-08-16
>
> > *The authors should consider adding a comparison of the differences between this work and other visual question answering datasets.*
>
> Visual Riddles is designed to test visual comprehension and factual grounding, focusing on how subtle visual details can significantly change understanding. Unlike typical VQA datasets, ours uniquely employs generated images based on real-world scenarios, enhanced with subtle clues that clarify otherwise ambiguous answers. This enables for more controlled and varied scenarios, distinguishing our dataset.
>
> Although many VQA datasets exist—ranging from synthetic images depicting unreal scenarios (like WHOOPS! [1]) to complex questions on real images that limit creative and ambiguous questioning (such as Sherlock [2], VisIT-Bench [3], and Encyclopedic-VQA [4])—our dataset's unique focus on fine-grained image understanding with commonsense and world knowledge makes direct comparison difficult. We discuss related datasets and benchmarks in Section 6, "Commonsense Reasoning in Multimodal Models" to highlight how our work contrasts with existing efforts. We will make sure it is clearer in the camera ready version.
>
>
>
> [1] Bitton-Guetta, N., Bitton, Y., Hessel, J., Schmidt, L., Elovici, Y., Stanovsky, G., & Schwartz, R. (2023). Breaking common sense: Whoops! a vision-and-language benchmark of synthetic and compositional images. In Proceedings of the IEEE/CVF International Conference on Computer Vision (pp. 2616-2627).
> ‏
>
> [2] Jack Hessel, Jena D Hwang, Jae Sung Park, Rowan Zellers, Chandra Bhagavatula, Anna Rohrbach, Kate Saenko, and Yejin Choi. The abduction of sherlock holmes: A dataset for visual abductive reasoning. In Computer Vision–ECCV 2022: 17th European Conference, Tel Aviv, Israel, October 23–27, 2022, Proceedings, Part XXXVI, pages 558–575. Springer, 2022.
>
>
> [3] Bitton, Y., Bansal, H., Hessel, J., Shao, R., Zhu, W., Awadalla, A., ... & Schmidt, L. (2023). Visit-bench: A dynamic benchmark for evaluating instruction-following vision-and-language models. Advances in Neural Information Processing Systems, 36, 26898-26922.
> ‏
>
> [4] Mensink, T., Uijlings, J., Castrejon, L., Goel, A., Cadar, F., Zhou, H., ... & Ferrari, V. (2023). Encyclopedic VQA: Visual questions about detailed properties of fine-grained categories. In Proceedings of the IEEE/CVF International Conference on Computer Vision (pp. 3113-3124).‏
>
> > *I think it's great that experienced designers are involved in building the dataset, but the size of the dataset still seems too small. It is also worth discussing whether the fine-tuning of the existing open source model can help improve the effect of the multi-modal large model in this benchmark.*
>
> Thank you for acknowledging the contribution of experienced designers to our dataset and raising concerns about its size. It’s important to note that recent vision-and-language evaluation efforts have increasingly focused on "quality over quantity" when evaluating foundation models. For example, datasets like Winoground [1] (CVPR 2022), with only 400 examples, have significantly influenced vision-language model development. Similarly, widely used datasets such as WHOOPS! [2] (ICCV 2023), LlaVA-Bench [3] (NeurIPS 2023), Visit-Bench [4] (NeurIPS 2024),ConTextual [5] (ICML 2024) and VibeEval [6] include 90, 500, 576, 500 and 269 examples respectively, and are popular in VLM evaluation. As you’ve highlighted and in line with this trend, while our dataset may appear small, it is important to understand that it is a high-quality, hand-crafted challenge set, designed specifically for testing the capabilities of multi-modal large models, not for training or fine-tuning. This distinction is crucial for understanding the dataset’s scope and role in the benchmark. Additionally, future work could explore automating dataset generation, developing a dedicated training set, using a model for this purpose, and leveraging Visual Riddles as a test set to further validate model performance.
>
> [1] Thrush, T., Jiang, R., Bartolo, M., Singh, A., Williams, A., Kiela, D., & Ross, C. (2022). Winoground: Probing vision and language models for visio-linguistic compositionality. In Proceedings of the IEEE/CVF Conference on Computer Vision and Pattern Recognition (pp. 5238-5248).
>
> [2] Bitton-Guetta, N., Bitton, Y., Hessel, J., Schmidt, L., Elovici, Y., Stanovsky, G., & Schwartz, R. (2023). Breaking common sense: Whoops! a vision-and-language benchmark of synthetic and compositional images. In Proceedings of the IEEE/CVF International Conference on Computer Vision (pp. 2616-2627).
>
> [3] Liu, H., Li, C., Wu, Q., & Lee, Y. J. (2024). Visual instruction tuning. Advances in neural information processing systems, 36.
> ‏
>
> [4] Bitton, Y., Bansal, H., Hessel, J., Shao, R., Zhu, W., Awadalla, A., ... & Schmidt, L. (2023). Visit-bench: A dynamic benchmark for evaluating instruction-following vision-and-language models. Advances in Neural Information Processing Systems, 36, 26898-26922.
> ‏
>
> [5] Wadhawan, R., Bansal, H., Chang, K. W., & Peng, N. (2024). ConTextual: Evaluating Context-Sensitive Text-Rich Visual Reasoning in Large Multimodal Models. arXiv preprint arXiv:2401.13311.
> ‏
> [6] Padlewski, P., Bain, M., Henderson, M., Zhu, Z., Relan, N., Pham, H., ... & Tay, Y. (2024). Vibe-Eval: A hard evaluation suite for measuring progress of multimodal language models. arXiv preprint arXiv:2405.02287.‏
>
>
> > *Figure 2.5: What is the meaning of CA and GTA?*
>
> *"CA"* stands for Candidate Answer, and *"GTA"* denotes Ground Truth Answer. We appreciate your attention to this detail and we added this information to the caption of Figure 2 (.5) and it will be visible in the camera-ready version of our paper.

---

> > ### Author Response · Authors · 2024-08-22
> >
> > Dear Reviewer g9pM,
> >
> > Thank you again for your constructive feedback. As the discussion period draws to a close in one week, we sincerely welcome any further feedback you may have. We deeply appreciate the time and effort you have already dedicated to reviewing our work.
> >
> > Sincerely,
> >
> > The Authors

---

### Official Review · Reviewer_WVhG · 2024-08-04
**Useful benchmark but further analysis would be helpful**

**Rating:** 7
**Confidence:** 3

**Review:**

Overall, the quality of the work appears good, but more clarity on the methodology and ultimate claims, analysis, and interpretation is necessary to help readers draw the appropriate conclusions from the benchmark results. Though not critical for providing benefit, further analysis would also be useful to show how this dataset builds on prior work.

Pros:
* Proposal of new dataset and benchmark, with more complex and nuanced visual cues and generated images, to evaluate the capabilities and limitations of current LVLMs.
* Care and attention toward process of building dataset, including multiple validated annotators and quality assurance for results
* Varied suite of evaluation tasks around dataset of visual riddles, including for free-response VQA and evaluation of proposed answers.
* Useful comparison against human results

Cons:
* Further details on the dataset and benchmark generation process and analysis of results needed to better understand how to interpret the results and where/why the LVLMs are failing
* Additional quantitative analysis required to demonstrate the necessity for this dataset, given that prior datasets do exist which similarly test comprehension and logical reasoning around visual cues and commonsense knowledge

**Strengths:**

This work demonstrates care toward building a robust dataset for leveraging visual cues and commonsense knowledge in VQA, with attention to annotator experience and quality assurance. The wide variety of benchmark tests is also important for understanding how well LVLMs perform under different scenarios, toward understanding the scenarios and reasons for why LVLMs perform poorly on this task. Overall, attention toward identifying the current capabilities and shortcomings of LLMs and LVLMs is a useful endeavor, and this benchmark will be useful in allowing for further studies of their capabilities.

**Additional Feedback:**

(8/21: edited to fix formatting issue)

(8/22: edited to change rating to 7)

**Clarity:**

See the "opportunities for improvement" section. Some parts of the process for building the dataset and human evaluation benchmark could be further clarified to help users understand their robustness.

**Correctness:**

LVLM and LLM models are used both for dataset generation and as the subjects of evaluation. While many LVLM models are used, I wonder if there is potential bias resulting from the choice of LVLM to generate images and evaluate captions. In particular, Gemini is one of the models used for generating the images as well as one of the models evaluated, and it happens to also be the one which achieved the best performance among LVLMs. Further clarification could be useful to justify any lack of bias here between models used in each use case.

See also prior points made about the method for human evaluation. While it is not necessarily incorrect, precision of the claim about what the human benchmark represents and how the process to obtain the mturk results accurately represents that benchmark would be important to ensuring that this comparison is correct.

**Documentation:**

To the best of my knowledge, all documentation and organization appears in order.

**Ethics:**

No significant ethical concerns.

**Limitations:**

The authors have identified limitations around the size of the dataset and potential biases or unethical depictions in the generated images. One other potential limitation that could be worth addressing is the practicality of how visual cues are typically leveraged in real life. While this is a task that humans can perform, it is also true that in most cases, humans are not analyzing a single image in a vacuum without other context/metadata, sensory inputs, and more. While this does not detract from what the benchmark can ultimately demonstrate, it may be worth noting this limitation or at least asserting why this benchmark is still important in spite of the limitation.

**Opportunities For Improvement:**

The main opportunities for improvement for me are around clarity of process and benchmarks, toward being able to draw justifiable conclusions around why the LVLMs are performing the way they are.

Some parts of the process for building the dataset and benchmarking performance could be further elaborated on. In particular, the paper and supplementary mention designers _experienced_ in generating images for building the dataset and _highly-rated_ mechanical turk workers but not the criteria by which this experience was evaluated. Understanding this would help understand better the quality claims of the authors and the reliability of the benchmark as well as how similar quality could be reproduced. Furthermore, one of the evaluation modes is combining human-generated captions with LLMs, but few details are given regarding the process of generating those captions. It seems like it would be important to understand the nature of the prompting and contents of the captions to understand the power of the LLMs versus the captions themselves. If the captions do not encode the important visual cues in text, then it seems likely that the LLM would be unable to effectively answer the question. This information seems important to be able to draw useful conclusions about the increased accuracy of this mode compared to using an LVLM end-to-end.

While the number of different evaluation tasks were appreciated, some further ablation studies could help with understanding why the LVLMs are achieving sub-human performance. I would have liked to see an ablation study identifying whether the models know the underlying common sense knowledge but are unable to string together in reasoning, or if they just aren't able to pull the commonsense knowledge needed to solve these problems. Further information about the caption experiments could also be useful toward understand this. More generally, it is not clear how well this benchmark actually extends to a useful application or conclusion about what LLMs can or can't do. I would have liked to see more analysis of why LVLMs are unable to perform as well on this task, more than just identifying that they can't perform at the same level of "human performance".

Lastly, I think it would be important to clarify precisely what the human evaluation is intended to represent. Because "high-quality" mechanical turkers are used to establish the human performance, with averaging of results, the interpretation of the benchmark differs from what we might describe as the average human performance. Furthermore, the annotators were allowed and advised to use other tools such as Google Lens. To that end, it feels important to clarify what exactly the human answers represent, and therefore what it means that the LVLMs do not perform to that level.

**Relation To Prior Work:**

This work is compared to other similar prior works such as VisIT-Bench and Encyclopedic VQA. The main contribution compared to these works is the production of new images not used in pretraining and the creation of more challenging scenarios. However, no work is referenced or produced here to demonstrate why these benchmarks are no longer sufficient. One would expect, if the images have now been seen during pretraining, that the current performance should reflect some degree of improved performance while new images with similar descriptions still demonstrate lower performance. More justification would be useful to contrast this work with prior work and to justify the need for this new benchmark.

**Summary And Contributions:**

This work proposes a new VQA-based benchmark for evaluating LVLM performance based on combining identification of subtle visual cues with leveraging commonsense knowledge to answer questions about a visual scenario.

Contributions include the following:
* Dataset of 400 visual riddles, which include a generated image and prompt requiring the user to use visual cues in the image as well as commonsense knowledge to answer the question
* Benchmark utilizing various scenarios to assess the model's VQA capabilities on the dataset, including in free-form, multiple choice, and evaluation settings
* Evaluation comparing several LVLM models to human performance on benchmark

---

> ### Author Rebuttal · Authors · 2024-08-16
>
> ### **Response - Part 1/2**
>
> Thank you for your valuable feedback on our paper. We appreciate your recognition of our dataset, benchmark, and evaluation methods, as well as your questions and suggestions on dataset generation and human captions. We will address these points to enhance the clarity and impact of our work.
>
>
> > *the paper and supplementary mention designers experienced in generating images for building the dataset and highly-rated mechanical turk workers but not the criteria by which this experience was evaluated. Understanding this would help understand better the quality claims of the authors and the reliability of the benchmark as well as how similar quality could be reproduced.*
>
> To confirm that the designers had the necessary skills, we conducted a trial where they were asked to successfully create 2-3 visual riddles based on examples from the provided guidelines. The designers are familiar with text-to-image models, and some have experience with vision-and-language datasets. Similarly, to ensure the raters (AMT workers who served as annotators for different tasks) were qualified, we administered a test requiring them to accurately evaluate 10 visual riddles before starting their annotation tasks. The designer guidelines will be included in the Appendix of the revised camera-ready version and detailed in Section 2 ("Data Collection") and A.2 of the Appendix. Guidelines for Mechanical Turk workers can be found in Sections A.5 and A.6 of the Appendix. The process of creating and annotating visual riddles can be replicated, but the challenge lies in crafting real-world scenarios that rely on subtle visual clues to avoid ambiguity.
>
> > *one of the evaluation modes is combining human-generated captions with LLMs, but few details are given regarding the process of generating those captions. It seems like it would be important to understand the nature of the prompting and contents of the captions to understand the power of the LLMs versus the captions themselves. If the captions do not encode the important visual cues in text, then it seems likely that the LLM would be unable to effectively answer the question. This information seems important to be able to draw useful conclusions about the increased accuracy of this mode compared to using an LVLM end-to-end.*
>
> To address this point, we conducted a new analysis comparing human (ground truth) captions with model (predicted) captions. The human captions are the original prompts our riddle creators gave to the text-to-image models to generate the visual riddle images, therefore these captions are "Oracle" in the sense that they typically contain the key elements needed to answer the riddle. We annotated 50% of the dataset to determine if each caption type included all the necessary details to solve the riddle. For example, in a riddle with a 1000-piece Dora the Explorer jigsaw puzzle asking, *"Should I buy this for my toddler?"*, the model-generated caption omitted the crucial **"1000-pieces"** detail, which is essential because such a puzzle is too complex for toddlers who typically use puzzles with fewer pieces. This omission made the question unanswerable.
> We found that 97% of human captions contained the required information, while only 57% of model-generated captions did. This discrepancy highlights the fact that questions pertain to small details in the image that models are not likely to "notice” when captioning. This analysis and examples of captions will be added to the camera-ready version and can be seen in the attached PDF in Figure 3.
>
> > *While the number of different evaluation tasks were appreciated, some further ablation studies could help with understanding why the LVLMs are achieving sub-human performance. I would have liked to see an ablation study identifying whether the models know the underlying common sense knowledge but are unable to string together in reasoning, or if they just aren't able to pull the commonsense knowledge needed to solve these problems.*
>
> > *Further information about the caption experiments could also be useful toward understand this.*
>
> We agree it is interesting and important to understand the performance of current models on our benchmark, and in particular what is the role of understanding common-sense. The paper actually has an ablation that is aimed at this question: the Caption-to-LLM experiment in Section 4.2, "Open-ended VQA". This ablation shows that even with Oracle human captions (as found in the analysis conducted above, 97% of which contain all necessary information), models still struggle with reasoning, achieving only a 50% success rate compared to 23% with model-generated captions. This highlights a 27% "visual recognition gap" (50% - 23%) where models could improve to oracle vision capabilities, and a 32% "reasoning gap" (82% - 50%) showing how much better a model with oracle vision ability could improve in reasoning compared to humans. We will clarify this in the paper.
>
> To address the last two points raised by the reviewer, both analyses underscore that the difficulty in solving our riddles arises from recognizing visual clues and reasoning over the provided information in relation to the question.

---

> > ### Author Rebuttal · Authors · 2024-08-16
> >
> > ### **Response - Part 2/2**
> >
> > > *More generally, it is not clear how well this benchmark actually extends to a useful application or conclusion about what LLMs can or can't do. I would have liked to see more analysis of why LVLMs are unable to perform as well on this task, more than just identifying that they can't perform at the same level of "human performance".*
> >
> > Thank you for highlighting the need to clarify the connection between our benchmark results and their practical implications for LLM capabilities. Our study does not merely establish that models fall short of human performance—it introduces challenging data that requires multi-step problem-solving. As noted above, even when provided with all necessary information (as in our Caption-to-LLM setup, Section 4.2 - "Open-ended VQA"), models struggle not only to recognize visual clues but also to apply reasoning and commonsense abilities.
> > We also examined if hints and attributions could improve performance. As shown in Section 4.4, hints slightly helped by directing models to relevant visual clues, but they didn't significantly enhance reasoning abilities. Attributions often caused confusion with irrelevant details, further illustrating the difficulty models have in visual reasoning.
> > We neutralized typical model responses like "cannot determine," which often arise from insufficient information, and observed how auxiliary inputs mitigated this tendency. Our ablation studies also highlight the models' struggles to utilize visual clues effectively and affirm the value of the carefully curated prompts and captions in our dataset, which generative models find challenging to replicate, especially with subtle visual cues. These insights collectively enhance our understanding of the current limitations and areas for improvement in LLMs' visual reasoning capacities.
> >
> > > *Lastly, I think it would be important to clarify precisely what the human evaluation is intended to represent. Because "high-quality" mechanical turkers are used to establish the human performance, with averaging of results, the interpretation of the benchmark differs from what we might describe as the average human performance. Furthermore, the annotators were allowed and advised to use other tools such as Google Lens. To that end, it feels important to clarify what exactly the human answers represent, and therefore what it means that the LVLMs do not perform to that level.*
> >
> > The human evaluation aims to set an upper limit for average human performance in solving the task. We selected annotators that passed the qualification test, which included all AM Turkers who demonstrated a clear understanding of how to solve open-ended VQA riddles and passed the qualification. No Turkers who passed were excluded from the annotation task. It's crucial to note that no professional skills are necessary; the primary requirement is grasping the fundamental concept of the visual riddles, where the key to solving lies in visual clues within the image. Thus, the Turkers are simply reliable annotators confirmed to comprehend the task.
> > Solving our visual riddles relies on several cognitive skills that are not traditionally tested in visual benchmarks. The integration of reasoning and the application of external knowledge are key elements that distinguish human cognitive abilities. Recognizing that advanced LLMs like Gemini and GPT have access to extensive datasets potentially surpassing average human knowledge, we permitted the use of auxiliary tools like Google Lens for our human annotators. For instance, Figure 2.4 (Multiple Choice) in the paper might benefit from using Google Lens for those unfamiliar with the Kakapo bird, whereas models may recognize the bird more easily. However, even without knowing its name, people can infer that something about the bird in the image hints at the solution and use the tool to identify it.
> > In response to your comment, we added a new experiment where we examine all riddles that potentially require tools like Google Search or Lens (those with attributions) and found that 59% don’t require them, while 41% may potentially require their use. However, model performance shows only a slight difference: 21% success rate when tools might be needed (because the solver does not have the world knowledge needed for solving the riddle) versus 26% when they are not, suggesting that tool usage is not the primary challenge for models. We will include this experiment in the camera ready version.

---

> > ### Comment · Reviewer_WVhG · 2024-08-21
> >
> > Thank you to the authors for the thorough responses to my concerns. I appreciate the additions and will keep my rating as is (7 - acceptance).

---

> > > ### Author Response · Authors · 2024-08-22
> > >
> > > Thank you for your thoughtful and constructive feedback on our rebuttal. We are grateful for your recognition of the improvements made to our paper. We noticed that your recent comment reflects a positive assessment of our work, suggesting you might have intended to rate it as 7 ("acceptance").
> > >
> > > If possible, could you please adjust the rating from 6 to 7 to align with your intended assessment? we greatly appreciate your consideration and the time you’ve dedicated to reviewing our work.

---

> > > > ### Comment · Reviewer_WVhG · 2024-08-22
> > > >
> > > > Thank you for pointing that out. I think it might have been a bug in the system, since I definitely double checked when submitting my comment. It should hopefully be fixed now, but please let me know if it still doesn't show that way on your end.

---

### Author Rebuttal · Authors · 2024-08-16

We appreciate the reviewers for recognizing the contributions of the Visual Riddles Benchmark and for their thoughtful feedback. The novelty of our dataset, particularly in combining subtle visual cues with commonsense knowledge, was well received (WVhG), as was the care put into its creation and the comprehensive evaluation of large vision-language models alongside human performance (5gUA). Reviewers also appreciated the clarity and originality of our approach in addressing commonsense reasoning and world knowledge (g9pM, zua6). In response to the valuable suggestions, we have added several analyses and figures to our paper, all of which can be found in the attached PDF file:
- Comparison of model-generated captions vs. ground-truth (human) captions and their impact on solving tasks.
- Aggregated results for model and human performance across difficulty levels and categories.
- Additional examples of visual riddles across different difficulty levels.


We are grateful for this recognition and address each reviewer’s comments in detail below.

---

### Decision · Program_Chairs · 2024-09-26

**Decision:**

Accept (Poster)

**Comment:**

The paper proposes Visual Riddles, a VQA dataset designed to evaluate the ability of vision language models to answer questions requiring a combination of visual reasoning and common sense/world knowledge.  The questions are designed to be ambiguous and potentially tricky to answer.  The dataset is hand-designed and consists of 400 images (synthetically generated with a text-to-image model) and paired with a textual question and hint to form a "visual riddle".  Different LVLM and LLM (with captions) models are benchmarked in several different scenarios (open ended answers, multiple choice, automated vs human evaluation).  Evaluation results show that there is still a substantial gap between human and LVLM/LLM performance (for the models that were evaluated).

All reviewers are positive on the work, finding the dataset to be carefully constructed and interesting.

The AC agrees that the dataset would be of value to the community as it provides a test bed for LVLM's ability to answer challenging questions.  However, the manuscript is lacking some details on the dataset as well as comparison to other VQA datasets.  The authors are encouraged to revise the paper in include:
1. Additional details on the dataset and benchmark generation (WVhG,5gUA)
   - Perhaps clarify in the main text a bit more detail of what is in the Appendix
2. More discussion of how this dataset relates to prior VQA datasets (WVhG, g9pM, zua6)
   - Please see Encyclopedic VQA [1] for related VQA datasets (including [2,3,4]) and examples of dataset comparison (e.g. through examples, statistics, etc).
   - The above datasets as well as other datasets (such as those provided in response to R-g9pM) should be discussed more in the Related Work section.
   - Following R-zua6 - please also move the Related Work section to be after the introduction.
3. Comparison of model performance on this dataset vs some representative prior VQA dataset (WVhG).  For instance, comparisons of performances of the evaluated LLM/LVLM on some selected VQA datasets.
4. Additional analysis of results (WVhG,5gUA)
   - Clarifications and additional analysis provided in author response
   - Statistical analysis on the results / elo ratings / aggregate evaluations over multiple seeds as suggested by R-5gUA

References:
[1] Encyclopedic VQA: Visual questions about detailed properties of fine-grained categories [Mensink et al. ICCV 2023]
[2] A-OKVQA: A Benchmark for Visual Question Answering using World Knowledge [Schwenk et al. ECCV 2022]
[3] Select, Substitute, Search: A New Benchmark for Knowledge-Augmented Visual Question Answering [Jain et al. SIGIR 2021]
[4] OK-VQA: A Visual Question Answering Benchmark Requiring External Knowledge [Marino et al. CVPR 2019]